



# Comparison of ocean vertical mixing schemes in the Max Planck Institute Earth System Model (MPI-ESM1.2)

Oliver Gutjahr[1], Nils Brüggemann[1,2], Helmuth Haak[1], Johann H. Jungclaus[1], Dian A. Putrasahan[1], Katja Lohmann[1], and Jin-Song von Storch[1,3]

[1]Max Planck Institute for Meteorology, Hamburg, Germany
[2]University of Hamburg, Hamburg, Germany
[3]Center for Earth System Research and Sustainability (CEN), University of Hamburg, Germany

**Correspondence:** O. Gutjahr (oliver.gutjahr@mpimet.mpg.de)

**Abstract.** We compare the effects of four different ocean vertical mixing schemes on the ocean mean state simulated by the Max Planck Institute Earth System Model (MPI-ESM1.2) in the framework of the Community Vertical Mixing (CVMix) library. Besides the PP and KPP scheme, we implemented the TKE scheme and a recently developed prognostic scheme for internal wave energy and its dissipation (IDEMIX) to replace the often assumed constant background diffusivity in the ocean

interior. We analyse in particular the effects of IDEMIX on the ocean mean state, when combined with TKE (TKE+IDEMIX).

In general, we find little sensitivity of the ocean surface, but considerable effects for the interior ocean. Overall, we cannot classify any scheme as superior, because they modify biases that vary by region or variable, but produce a similar pattern on the global scale.

However, using a more realistic and energetically consistent scheme (TKE+IDEMIX) produces a more heterogeneous pattern

of vertical diffusion, with lower diffusivity in deep and flat-bottom basins and elevated turbulence over rough topography. In addition, TKE+IDEMIX improves the circulation in the Nordic Seas and Fram Strait, thus reducing the warm bias of the Atlantic water (AW) layer in the Arctic Ocean to a similar extent as has been demonstrated with eddy-resolving ocean models.

We conclude that although shortcomings due to model resolution determine the global-scale bias pattern, the choice of the vertical mixing scheme may play an important role for regional biases.

## 1   Introduction

Vertical mixing in the ocean is a complex phenomena and its magnitude depends on processes acting over a large range of vertical and horizontal scales, from about 1 km to several meters down to centimetres (Fox-Kemper et al., 2019). Vertical mixing affects key elements in the ocean that are of climatic importance, such as ocean stratification, the distribution of passive

tracers such as temperature and salinity, the uptake of heat and carbon, and the global meridional overturning circulation (MOC; e.g. Gent, 2018).



In ocean models, the processes that lead to mixing are subgrid-scale and therefore not resolved, so they have to be parameterised. The complexity of these parameterisations varies in dependence of our understanding (e.g. Large et al., 1994; Fox-Kemper et al., 2019). In fact, the parameterisation of vertical mixing constitutes one of the current shortcomings of ocean
models (Robertson and Dong, 2019; Fox-Kemper et al., 2019).

Here, we aim to compare four state-of-the-art ocean vertical mixing schemes in coupled simulations with the Max Planck Institute Earth System Model (MPI-ESM1.2). Such a comparison provides a better understanding of the model behaviour at the process level, especially when the same model is used. It is therefore very helpful to have libraries that provide different choices for parameterisations, such as the Community Vertical Mixing (CVMix) library (Griffies et al., 2015; Van Roekel et al.,
2018), which we have coupled to MPI-ESM1.2 and extended by two additional schemes (see section 2 and appendix A3).

Frequently used ocean vertical mixing schemes date back to the 1980s and 1990s. Often a modelling centre or a group decides to implement only one of these schemes and, for practical reasons, not to deviate from it afterwards. However, as these schemes are based on different principles, deviations in the results are to be expected.

In the ocean surface boundary layer, schemes provide either direct vertical profiles of scalar mixing diffusivity and viscosity,
such as in the PP scheme (Pacanowski and Philander, 1981) or in the K-profile parameterisation (KPP; Large et al., 1994), or fluxes and covariances in the case of second-order schemes (Mellor and Yamada, 1982), such as in the turbulent kinetic energy (TKE) scheme (Gaspar et al., 1990). These two most common approaches represent processes that result in vertical shear of the velocity and in changes of the buoyancy, e.g. due to convection. These schemes can become more complex by adding important sub-grid scale processes (Fox-Kemper et al., 2019), such as mixing by Langmuir turbulence (e.g. McWilliams et al.,
1997; Li and Fox-Kemper, 2017; Li et al., 2019) or internal tides (Garrett, 2003).

The default scheme in MPI-ESM1.2 is a modified version of the formulation of Pacanowski and Philander (1981) (PP). However, recent experiments with ECHAM6.3 at T255 resolution for the atmosphere, resulted in unstable AMOC states and in a freezing of the Labrador Sea (Putrasahan et al., 2019). Replacing the PP scheme with the K-profile parameterisation (KPP; Large et al., 1994) resolved this issue, maintaining a stable AMOC (Gutjahr et al., 2019). We perform additional sensitivity
experiments in which we replace the PP scheme by a scheme based on turbulent kinetic energy (Gaspar et al., 1990), to which we refer as the TKE scheme. See a detailed description for all used vertical mixing schemes in appendix A.

The background diffusivity, which quantifies the mixing due to internal wave breaking, is either parameterised as a constant value (PP, KPP) or it depends on the buoyancy frequency and an artificial minimum value for the turbulent kinetic energy (TKE scheme). We have implemented the TKE scheme with two alternatives for the background diffusivity. In the first case
we use a minimum value for TKE, which represents the breaking of internal waves. In the second case, we combine the TKE scheme with IDEMIX (Internal Wave Dissipation, Energy and Mixing), which describes the energy transfer from internal wave sources to wave sinks using the radiative transfer equation of weakly interacting internal waves (Olbers and Eden, 2013; Eden et al., 2014). Energy that is dissipated by internal waves is treated as an energy source of turbulent kinetic energy, leading to an energetically more consistent solution (Eden et al., 2014).

Furthermore, IDEMIX solves the propagation of low-mode waves away from their generation site (Fox-Kemper et al., 2019) (see appendix section A3 and Fig. A1), along with the energy loss the waves experience as they encounter different ocean



regions and continental shelves. Compared to empirical tidal mixing schemes, e.g. Simmons et al. (2004), IDEMIX not only represents internal waves generated by barotropic tides interacting with rough topography, but also internal waves excited at the base of the mixed layer due to high-frequency wind fluctuations.

IDEMIX has been developed recently and its performance was studied in both stand-alone ocean models (Eden et al., 2014; Pollmann et al., 2017; Nielsen et al., 2018) and coupled simulations (Nielsen et al., 2018, 2019). Based on ocean-only simulations, the TKE dissipation calculated with a combined TKE and IDEMIX scheme agrees well with Argo float-derived dissipation rates (Pollmann et al., 2017). Using IDEMIX in coupled simulations by Nielsen et al. (2018) revealed only a minor effect on the climate state. However, they demonstrate reduced thermocline diffusivities with IDEMIX, which leads to a sharper

and shallower thermocline, because less heat is mixed downwards.

Despite the latter but because of the promising results with respect to results from Pollmann et al. (2017) and Eden et al. (2014), we compare the effect of IDEMIX with the other mixing schemes of MPI-ESM1.2 and analyse regions that are most sensitive to IDEMIX on the typical time scale of 100 years for climate simulations.

The remainder of the manuscript is organized as follows. In section 2 we briefly describe the model and experiments, the
results of the comparison in section 3 and 4, and we conclude in section 5.

## 2   Model description

For our analysis we used the Max Planck Institute Earth System Model in version 1.2.01 (MPI-ESM1.2; Mauritsen et al., 2019). The model consists of the atmospheric submodel ECHAM6.3 (Stevens et al., 2013), including the land-surface submodel JSBACH3.2, and of the ice-ocean submodel MPIOM1.6.3 (Jungclaus et al., 2013; Notz et al., 2013). The submodels are
coupled via the Ocean-Atmosphere-Sea-Ice coupler version 3 (OASIS3-mct; Valcke, 2013) with a coupling frequency of 1 h.

The horizontal resolution of the atmosphere is T127 (about 103 km) with 95 vertical levels. The ocean is discretised on a tripolar grid with a horizontal resolution of $0.4°$ (TP04; about 44 km) and 40 vertical levels, of which the first 20 levels are distributed in the top 750 m. A partial grid cell formulation (Adcroft et al., 1997; Wolff et al., 1997) was used to better represent the bottom topography. River runoff was calculated by a horizontal discharge model (Hagemann and Gates, 2003). Tracer
advection by unresolved mesoscale eddies is parameterised following Gent et al. (1995) (GM) with a constant eddy thickness diffusivity of $250\,\mathrm{m^2\,s^{-1}}$ for a 400 km wide grid cell, which reduces linearly with increasing resolution (about $25\,\mathrm{m^2\,s^{-1}}$ for a resolution of 40 km). The lateral eddy diffusivity is parameterised by an isopycnal formulation (Redi, 1982) with a constant value of $1000\,\mathrm{m^2\,s^{-1}}$ for a 400 km wide grid cell, which again reduces linearly with increasing resolution (about $100\,\mathrm{m^2\,s^{-1}}$ for a resolution of 40 km).

The default parameterisation of ocean vertical mixing is a modified version of the Pacanowski and Philander (1981) or PP scheme, extended with a wind-induced mixing term (Marsland et al., 2003) (see Appendix A for details). The PP scheme was also used to tune MPI-ESM1.2-HR (Mauritsen et al., 2012). This model configuration is referred to as "high resolution" (HR) and was described and tested in more detail by Mauritsen et al. (2019), Müller et al. (2018), and Gutjahr et al. (2019).



To analyse the effect of different ocean vertical mixing schemes on the ocean mean state, we coupled the Community Vertical
Mixing (CVMix) library (Griffies et al., 2015; Van Roekel et al., 2018) to MPI-ESM1.2. The K-profile parameterisation (KPP)
as described in (Large et al., 1994) is already included in CVMix and has been used with MPI-ESM1.2 by Gutjahr et al. (2019).
In addition, we have added the (TKE; Gaspar et al., 1990; Eden et al., 2014) and IDEMIX (Olbers and Eden, 2013) to CVMix.
Although it is principally possible to couple IDEMIX to other mixing schemes, such as KPP, we only combined it with TKE,
because both schemes rely on energy budgets, which results in a consistent mixing scheme.

**2.1   Experiments**

We performed four 100-year long control simulations with MPI-ESM1.2-HR using four different ocean vertical mixing
schemes. See Tab. 1 for an overview of the experiments and Appendix A for details of the mixing schemes.
    The reference simulation ($HR_{pp}$) uses the PP scheme and exactly the configuration used by Müller et al. (2018) and Gutjahr
et al. (2019). In the second simulation we used the KPP scheme and refer to it as $HR_{kpp}$. The configuration of this experiment
is exactly as in Gutjahr et al. (2019). These two simulations were also compared with higher resolution versions (atmosphere
and ocean) by Gutjahr et al. (2019), as part of the High-Resolution Model Intercomparison Project (HighResMIP; Haarsma
et al., 2016).
    The third experiment ($HR_{tke}$) used the TKE scheme with a background diffusivity depending on the buoyancy frequency
and a minimum value for turbulent kinetic energy (see Appendix A3), but without any contribution from IDEMIX. In the last
experiment ($HR_{ide}$), we used the TKE scheme together with IDEMIX (TKE+IDEMIX) and replaced the artificial background
diffusivity with one diagnosed from turbulent kinetic energy that is fueled by the internal wave dissipation (see section A3 for
more details). If not explicitly mentioned, we used default values for the parameters of the mixing schemes, as listed in the
respective original description (see also Appendix A), without analysing the effect of changed parameters.
    The initial state was derived from a MPI-ESM1.2-HR simulation (with the PP scheme) that was nudged to the averaged
temperature and salinity state of 1950 to 1954 of the Met Office Hadley Centre EN4 observational data set (version 4.2.0;
Good et al. (2013)), as described in Gutjahr et al. (2019). All simulations were forced by constant 1950s conditions according
to the HighResMIP protocol (Haarsma et al., 2016). As recommended in that protocol, the model was not re-tuned to obtain
unbiased effects from the change of the ocean vertical mixing scheme. If not stated otherwise, we analysed averages over the
last 20 model years (model years 81 to 100). Furthermore, we restrict the comparisons to the ocean mean state, being aware
that there possibly are different responses and feedbacks with the atmosphere.

**3   Global comparison**

In the following, we present how the choice of the ocean vertical mixing scheme affects the ocean mean state in control
simulations with MPI-ESM1.2-HR. We first present results for the global ocean, before discussing specific regional aspects
(section 4).





### 3.1 Spatial distribution of the vertical diffusivity

For large parts of the ocean, the vertical diffusivity $K$ is homogeneously distributed in the first order, as demonstrated exemplary at intermediate depth of 1020 m (Fig. 1); taking on the background value $K = 1.05 \cdot 10^{-5}\,\mathrm{m^2\,s^{-2}}$ in the case of $\mathrm{HR_{pp}}$ and $\mathrm{HR_{kpp}}$. Because of the relationship $K = \sqrt{2}E_{\mathrm{min}}/N$ for the background diffusivity in the TKE scheme (see Appendix A3), $\mathrm{HR_{tke}}$ simulates a small $K$ in the tropical and subtropical ocean, where $N$ is large and a large $K$ in the high-latitude ocean, where $N$ is small. Even more heterogeneous is the distribution of $K$ in $\mathrm{HR_{ide}}$, which simulates stronger mixing above rough topography and mixing coefficients of about two orders of magnitude lower above the abyssal plains and in the Arctic Ocean. Hot spots of strong vertical mixing are simulated for all four cases particularly in the subpolar North Atlantic (SPNA), in the Nordic Seas, and in the Weddell and Ross Sea of the Southern Ocean. Excessive deep convection in the Weddell Sea is a known issue in ocean models (e.g. Sallée et al., 2013; Kjellsson et al., 2015; Heuzé et al., 2015; Naughten et al., 2018) and not unique to MPI-ESM1.2-HR. The unrealistic convection is related to anomalously frequent open-ocean Weddell Sea polynyas (Gordon, 1978; Carsey, 1980; Gordon, 2014). $\mathrm{HR_{ide}}$ reduce the occurrence of this spurious deep convection, which we will discuss in section 4.3.

A closer look at Fig. 1 reveals more regional differences in the above-mentioned areas, which we will discuss in the following. We will further relate these differences to the temperature and salinity bias (section 3.2 and 3.3 ) and discuss them in more detail for the Arctic Ocean (section 4.1), SPNA and Nordic Seas (section 4.2), and Southern Ocean (section 4.3).

### 3.2 Sea surface temperature and salinity bias

The sea surface temperature (SST) is a key quantity for the atmosphere-ocean coupling. However, ocean-only and coupled climate models generate biases, the causes of which are often complex. Among others, the resolution, discretisation, and parameterisation of subgrid-scale processes have the largest effect on the biases (Fox-Kemper et al., 2019), with vertical mixing being just one complex process.

Overall, the SST is mostly colder in our simulations compared with EN4 (Fig. 2), with some exceptions such as in upwelling regions, in the North Atlantic subpolar gyre, and along the Labrador Current. Although vertical mixing has minor influence on SST bias in large parts of the ocean, some areas react more sensitively. One such area is the North Atlantic with its subpolar gyre, as well as the Nordic Seas. The largest cold bias occurs in the North Atlantic and amounts to -7°C in $\mathrm{HR_{pp}}$. This cold bias is a common phenomena in ocean models (e.g. Randall et al., 2007) and is mainly caused by a too zonal pathway of the Gulf Stream (Dengg et al., 1996) in relation to insufficient horizontal resolution and northward heat transport (Wang et al., 2014). The North Atlantic SST is sensitive to the exchange of the vertical mixing scheme in that the cold bias is attenuated (Fig. 2b-d). However, this cold bias reduction comes at the expense of an even warmer bias in the subpolar gyre and Nordic Seas than in $\mathrm{HR_{pp}}$. Although the sea ice edge is in better agreement with observations (not shown), it is probably caused by the wrong reason. We will discuss these changes in more detail in section 4.2.


The sea surface salinity is mostly unaffected by the chosen vertical mixing scheme, except in the Arctic Ocean (Fig. 3). This bias is considerably reduced by using TKE or TKE+IDEMIX, as the surface waters become fresher, especially in the Canada Basin (Fig. 3c-d), probably due to increased inflow from the McKenzie River.

### 3.3 Ocean interior

#### 155 3.3.1 Horizontal maps of hydrographic biases

At intermediate depth, all simulations are too warm with respect to EN4, as shown in the temperature bias at 740 m depth (Fig. 4). Exceptions are the Southern Ocean and parts of the North Atlantic, where the ocean is colder at upper to intermediate depth. In the Atlantic Ocean, the warm biases are mainly linked to the representation of the Agulhas Current System and Mediterranean Overflow water (MOW), as well as to the pathway of the Gulf Stream. Previous studies with MPI-ESM1.2 have
shown that these biases diminish with increasing spatial resolution (Gutjahr et al., 2019; Putrasahan et al., 2019). Advection of these warmer (and more saline) water masses causes subsequent warm biases in the SPNA, Nordic Seas and Arctic Ocean. Even though an eddy-resolving resolution reduces most of these biases, as shown by Gutjahr et al. (2019) with MPI-ESM1.2-ER ($0.1°$), the choice of the vertical mixing scheme also affects the hydrographic biases, for instance in the Arctic Ocean and for the MOW warm bias (Fig. 4).

The salinity shows a similar bias pattern at intermediate depth (Fig. 5). The Atlantic is too saline, which is again due to the poor representation of the MOW and the Agulhas Current System. Consequently, northward advection by the Gulf Stream and the boundary currents along the European shelf distribute these saline waters into the subpolar North Atlantic and Nordic Seas. At a resolution of $0.4°$, our model is unable to capture the Agulhas Retroflection. Although all simulations show a similar salinity bias in the Agulhas region, we note a larger bias for $HR_{kpp}$, $HR_{tke}$, and $HR_{ide}$, which indicates a stronger inflow of
saline water from the Indian Ocean.

The largest difference in salinity bias is linked to the representation of MOW. Although all models produce warmer and more saline MOW, the bias is decreased only in $HR_{tke}$. The bias even becomes considerably larger when IDEMIX is used in the TKE scheme instead of an artificial background diffusivity (Fig. 5d). Probably, using IDEMIX reduces the vertical mixing in the Mediterranean Sea and especially near the overflow sill in the Strait of Gibraltar and downstream thereof. This lower
diffusivity reduces mixing with the ambient less saline water, so that the warm, highly saline core of the MOW is less diluted than in the other simulations.

We further note a slight freshwater bias in the Arctic Ocean in $HR_{ide}$ that we will describe in relation to the Atlantic water layer in section 4.1.

#### 3.3.2 Vertical section through the Atlantic and Arctic Ocean

A vertical section of the zonal averaged potential temperature bias through the Atlantic and Arctic Ocean (Fig. 6), shows again warm biases related to the above named currents and water masses. Salinity shows a similar bias pattern (not shown) with too saline waters where there is a warm bias.





The simulations show pronounced warm biases in the Atlantic water (AW) layer in the Arctic Ocean, in the Nordic Seas, for the overflow waters at roughly $60\,°$N, for the MOW at $30\,°$N at about 800 to $1000\,$m depth, and at roughly $40\,°$S in relation with Agulhas leakage and hence too much inflow from the Indian Ocean. $HR_{kpp}$ shows almost no difference to the reference simulation, except for slightly warmer overflow waters and a slightly warmer Arctic Ocean. A similar result is found for $HR_{tke}$, except for a lower MOW bias. In contrast, the warm bias of the AW layer in the Arctic Ocean is reduced in $HR_{ide}$, so that biases north of Svalbard are smallest in $HR_{ide}$, and the warm bias related to the MOW and Agulhas leakage become stronger.

## 4   Regional comparison

In the next sections we will discuss in more detail some of the above mentioned areas where IDEMIX influences the model bias both positively and negatively. We already note here that the insufficient model resolution determines the large-scale bias pattern, as shown by Gutjahr et al. (2019). However, the choice of the vertical mixing scheme plays an important role on the regional scale. In particular the Arctic Ocean, the subpolar North Atlantic and the Southern Ocean are sensitive to the chosen mixing scheme.

### 4.1   Atlantic water layer in the Arctic Ocean and the circulation at Fram Strait

#### 4.1.1   Fram Strait

A well-known error of many state-of-the-art ocean models is an anomalously thick and deep AW layer (e.g. Holloway et al., 2007; Shu et al., 2019). This error is thought to be related to model resolution and to vertical mixing schemes, in particular to the choice of the fixed background mixing value (Zhang and Steele, 2007; Liang and Losch, 2018). In terms of model resolution, it was recently shown that a high-resolution ocean ($1/10\,°$ or better) reduces this bias (Wang et al., 2018; Gutjahr et al., 2019). Here, however, we demonstrate that the bias is also reduced by using an improved vertical mixing scheme (i.e. TKE+IDEMIX), which produces a stronger southward recirculation at Fram Strait and stronger heat loss due to enhanced mixing.

Warm and saline AW is the main contributor of salt and oceanic heat into the Arctic Ocean. The AW flows northward first as the Norwegian Current in the Nordic Seas until it splits into two branches south of Svalbard (Fig. 7). A small branch enters the Barents Sea via the Barents Sea Opening (Barents Sea Water Branch) and flows into the Arctic Ocean at the St. Anna Trough in the northern Kara Sea. The main branch continues northward as the West Spitsbergen Current (WSC). About half of the AW transported in the WSC recirculates between 76 and $81\,°$N, thereby becoming part of the southward flowing East Greenland Current (EGC). Observations suggest the strongest recirculation at about $79\,°$N ('southern' recirculation; Marnela et al., 2013).

A 'northern' recirculation was observed from moorings at about $81\,°$N in association with the cyclonic gyre around the Molloy Hole ($79°8'12''$N; $2°49'0''$E) and mesoscale eddies (Marnela et al., 2013; Hattermann et al., 2016). Because of the small local Rossby radius of about 3 to $6\,$km in the WSC (von Appen et al., 2016) and about $6\,$km in the EGC (Zhao et al., 2014), most ocean models do not resolve eddies and thus simulate no 'northern' recirculation but a recirculation that is too





far south compared to observations (Fieg et al., 2010). However, several studies with eddy-permitting/resolving ocean models
(e.g. Wang et al., 2018; Fieg et al., 2010; Kawasaki and Hasumi, 2016) showed a 'northern' recirculation and eddy-activity in
the Fram Strait.

The other, non-recirculating half of the AW flows through Fram Strait into the Arctic Ocean (Fram Strait Branch Water),
supplying the AW layer at 200 to 700 m (Rudels et al., 1994), with its core at a depth at about 400 m.

The first indication why the AW layer warm bias in $HR_{ide}$ is smaller (Fig. 8) relates to a stronger recirculation in the Fram
Strait, which splits only in $HR_{ide}$ into a 'southern' (south of 79°N) and a 'northern' recirculation (north of 79°N). A stronger
recirculation means less inflow of AW and thus less heat transport into the Arctic Ocean. This stronger return circulation
seems to be related to enhanced deep convection in the Nordic Seas (Fig. 11), which in turn accelerates the geostrophic flow
around the convection centre due to steeper isopycnal gradients. In fact, the Greenland Sea gyre is up to 5 Sv stronger in $HR_{ide}$
(Fig. 12d). Furthermore, the 'northern' recirculation transfers volume and heat from the WSC to the EGC, which thus does not
enter the Arctic Ocean. A similar circulation structure develops in the eddy-resolving MPI-ESM1.2-ER model (Gutjahr et al.,
2019, not shown), but with sharper defined currents. In general, eddy-resolving models were found to improve the water mass
properties and the separation of the WSC into a recirculating branch (e.g. Hattermann et al., 2016; Wekerle et al., 2017). Note,
however, that the HR model does not resolve eddies in the Fram Strait due to the small Rossby radius at this high latitude.

The final passage of the WSC into the Arctic Ocean is at the Yermak Plateau (YP), a bathymetric feature north of Svalbard
known as a hot spot for internal wave activity and mixing (e.g. Fer et al., 2010; Crews et al., 2019). Here the diverging isobaths
split the WSC into three branches: 1) the Svalbard branch that flows along the upper inward continental slope of Svalbard
(above 500 m depth) 2) the Yermak branch (Crews et al., 2019), which is the lower boundary current along the margin of the
YP, roughly along the 1000 m-isobath, and 3) the recirculating branch between 78 and 80°N (Marnela et al., 2013) that flows
southward as part of the EGC. However, recent studies indicate a third pathway of the WSC, the Yermak Pass Branch, by which
a significant portion of the AW traverses the YP along the 700 m isobath and enters the Arctic Ocean (Koenig et al., 2017).

At the surface, the WSC looses heat directly to the relatively cold atmosphere, while at depth the heat loss is due to turbulent
mixing induced by currents and internal waves, and to eddy activity (Fer et al., 2010). Based on measurements, the cooling
strength depends on whether the WSC flows around or over the YP, resulting in different mixing rates (Fer et al., 2010).
Downstream the YP, the Svalbard and Yermak branches flow along the shelf break to the east, where they encounter the
Barents Sea Water Branch at the outlet of the St. Anna Trough (Dmitrenko et al., 2015). The St. Anna Trough is about 300 to
500 m deep in its southern part, reaching 1000 m at the entrance to the Arctic Ocean (Lien and Trofimov, 2013). Both water
masses continue thereafter as part of the cyclonic Arctic Circumpolar Boundary Current (Rudels et al., 1999; Aksenov et al.,
2011), although most of Fram Strait branch seems to recirculate already in the Nansen Basin (Rudels et al., 2015).

In $HR_{pp}$, $HR_{kpp}$, and $HR_{tke}$ the warm bias of the AW layer is about +2 to +3°C at YP and approximately +1 to +2°C
further downstream along the shelf break of the Eurasian Basin (Fig. 8). Some of the AW also crosses the Lomonosov Ridge
and spreads into the Markarov and Canada basins. In $HR_{ide}$ the AW is still too warm at the YP, but not further downstream into
the Arctic.





A comparison of $K$ at 450 m depth at the Fram Strait and in the Arctic Ocean (Fig. 9) reveals two important differences in HR$_{\text{ide}}$. First, the mixing near YP is slightly stronger in HR$_{\text{ide}}$ (Fig. 9d). This increased mixing at YP causes a stronger heat release into the atmosphere, which cools the AW more effectively than in the other simulations. In fact, the sensible heat flux is about 20 to $40\,\mathrm{W\,m^{-2}}$ larger in HR$_{\text{ide}}$ than in HR$_{\text{pp}}$ (not shown). For comparison, the sensible heat flux is only about 10 to $20\,\mathrm{W\,m^{-2}}$ stronger in HR$_{\text{kpp}}$ and HR$_{\text{tke}}$. However, the heat loss, especially the sensible heat flux, is already stronger in the Greenland Sea, further densifying the water for deep convection, and the temperature of the AW is already about $1\,^{\circ}\mathrm{C}$ colder at the Fram Strait in HR$_{\text{ide}}$. This stronger heat loss explains why the temperature of the AW at the Fram Strait is less in HR$_{\text{ide}}$, even though the Greenland Sea Gyre is stronger and thus warmer AW could be expected (Chatterjee et al., 2018; Muilwijk et al., 2019).

Along the shelf-break of the Nansen Basin, all simulations show a sharp band of high mixing coefficients aligned to bathymetry, where polar and shelf waters entrain into the cyclonically spreading AW. A local maximum of vertical mixing is further simulated at the outlet of the St. Anna Trough in the northern Kara Sea, where the Barents Sea Water Branch spills into the Nansen Basin and encounters the Fram Strait Branch Water.

### 4.1.2 Arctic Ocean

Turbulence in the quiescent interior Arctic Ocean is close to the limits of measurement (Rainville and Winsor, 2008), which are very rare and therefore mixing rates are largely unknown (Fer, 2009). The Arctic Ocean is isolated from the atmosphere by sea ice, such that wind stress is a minor source for mixing in all simulations. In the models, the wind stress thus reduces quadratically with the sea ice concentration (see Appendix A). In addition, brine rejection is less effective as a mixing mechanism because of the strong vertical salinity gradients in the cold halocline layer. Therefore, apart from enhanced mixing by episodic shear events, storms during ice-free conditions (Rainville and Woodgate, 2009), mesoscale eddies, or lateral intrusion along the boundaries, vertical diffusive mixing dominates over turbulent mixing (Fer, 2009).

Internal wave activity is almost absent, except above rough topographic features. In fact, internal waves are trapped at the place of their origin and do not propagate far into the Arctic Ocean. This trapping occurs because the Arctic Ocean is north of the critical latitude, which is $74.5\,^{\circ}\mathrm{N}$ for the M$_2$ tide, beyond which the Earth's rotation prohibits freely propagating internal waves. As a result, they dissipate at or very close to their source region with properties similar to lee waves (Rippeth et al., 2015, 2017).

For this reason, there is little or no contribution to small-scale turbulence in the inner Arctic Ocean in HR$_{\text{ide}}$, especially in the deep and flat-bottomed Canada basin. The mixing coefficient $K$ is up to two orders of magnitude smaller ($O(10^{-6}\,\mathrm{to}\,10^{-7}\,\mathrm{m^2\,s^{-1}})$) in HR$_{\text{ide}}$ compared to the other simulations, where $K$ is mostly at the constant background value ($1.05 \cdot 10^{-5}\,\mathrm{m^2\,s^{-1}}$). Representing this trapping of internal waves is crucial to simulate mixing rates that are more consistent with microstructure measurements, which show low turbulent diffusivity in deep, flat-bottomed basins, but elevated mixing rates above deep topography (Rainville and Winsor, 2008). The low diffusivities in deep basins agree well with observations from the Barneo ice camp drift, where $O(10^{-6}\,\mathrm{m^2\,s^{-1}})$ was measured below the mixed layer in the Amundsen Basin (Fer, 2009). A contrasting example of higher mixing rates in the inner Arctic Ocean is a distinct band of strong mixing along and above the Lomonosov





Ridge (Fig. 9d). Here, internal waves dissipate immediately after their formation, elevating turbulence locally, what was also measured directly by Rainville and Winsor (2008).

Assuming a constant background diffusivity in the Arctic Ocean thus largely overestimates the vertical mixing in the Arctic
Ocean. Although the background diffusivity can be artificially reduced in the Arctic Ocean to mimic this low internal wave activity (e.g. Kim et al., 2015; Sein et al., 2018), the very heterogeneous pattern described above is not captured. Coupling IDEMIX to TKE, however, is able to reproduce the spatial pattern and correct magnitudes. It further provides an energetically consistent solution.

Much lower vertical diffusivity under the sea ice in the Arctic Ocean might further cause denser water to enter the Nordic
Seas (Kim et al., 2015), pushing up the isopycnals and producing denser deep water, which then increased the overflow volume above the Greenland-Iceland-Scotland Ridge.

### 4.2 Subpolar North Atlantic and the Nordic Seas

#### 4.2.1 Convection and mixed layer depths

The subpolar North Atlantic (SPNA) and the Nordic Sea are important regions for the global climate, where the vertical
connection between the upper warm and the lower cold branch of the Atlantic meridional overturning circulation (AMOC) is established. The northward flowing warm AW is cooled by extensive heat loss to the atmosphere until it becomes dense enough to sink into deeper layers. Together with the dense overflow water from the Nordic Seas, it leaves the SPNA as North Atlantic Deep Water (NADW) with the southward-flowing Deep Western Boundary Current (DWBC).

The sinking of these dense water masses is thought to result from downwelling along the boundary current with complex
interplay of deep convection and the mesoscale eddy field (e.g. Katsman et al., 2018; Brüggemann and Katsman, 2019; Sayol et al., 2019; Georgiou et al., 2019). The SPNA and the Nordic Seas are one of the few places where deep convection of the open ocean occurs. Convection, or vertical instability, is parameterised differently in the vertical mixing schemes in MPI-ESM1.2 (see Appendix A), which is why we expect differences in vertical diffusion and mixed layer depths (MLDs).

All simulations generate the largest values of $K$ in the Labrador Sea and the Nordic Seas (Fig. 10). In the sensitivity
simulations (Fig. 10b-d) the diffusivity is markedly greater than in the reference simulation. In particular, we note an increased vertical diffusivity in the Irminger Sea, which has been recognized as an area where open-ocean deep convection takes place, contributing to the formation of Labrador Sea Water (e.g. Pickart et al., 2003; Våge et al., 2011).

In $\mathrm{HR_{pp}}$ the vertical instability is parameterised by enhancing the diffusivity to $K = 0.1\,\mathrm{m^2\,s^{-1}}$. The convection parameterisation is more complex in $\mathrm{HR_{kpp}}$, where non-local transport terms (see section A2) redistribute the surface fluxes throughout
the ocean surface boundary layer. These non-local transport terms depend on the net heat and freshwater fluxes at the ocean surface, on $K$, and on a dimensionless vertical shape function (Large et al., 1994; Griffies et al., 2015). In $\mathrm{HR_{tke}}$ and in $\mathrm{HR_{ide}}$, the buoyancy term (third term on the r.h.s. of Eq. A15), which usually is an energy transfer from TKE to mean potential energy, acts in this case ($N^2 < 0$) in opposite direction, thus enhancing TKE. However, besides differences in the parameterisations,





remotely changed water mass properties also affect convection and the MLD in the SPNA. Therefore, it is not straightforward
to diagnose what is a cause and what is a consequence for changes in the MLDs.

The average MLDs in March are shown in Fig. 11. A direct comparison with MLDs derived from Argo floats are not optimal,
because our simulations are control simulations with 1950s greenhouse gas forcing and we apply a different density threshold.
Keeping this in mind, we find profound differences to Argo float-derived MLDs and across the simulations. As with vertical
diffusivity, all simulations show the deepest mixed layers in the Labrador Sea and a second maximum in the GIN Seas. In
general, the sensitivity simulations all tend to simulate deeper mixed layers in the SPNA compared with $HR_{pp}$. In the Labrador
Sea, the deep mixed layers extend too far north in all simulations due to the lack of mesoscale eddies, which would impede
convection and restratify the water column (e.g. Eden and Böning, 2002; Brüggemann and Katsman, 2019; Gutjahr et al.,
2019).

$HR_{ide}$ simulates deeper mixed layers in the Greenland Sea, in particular around Jan Mayen, which might be caused by internal
wave activity, especially along the Kolbeinsey and Mohn Ridge and along the Jan Mayen Fracture Zone. The boundary current
in the GIN Seas is about $0.5\,°C$ colder in $HR_{ide}$ (not shown) and the anticyclonic Lofoten Basin Eddy (Yu et al., 2017), the area
of the strongest heat release, is about $1\,°C$ colder than in $HR_{pp}$, which constitutes a bias of about $-2\,°C$ compared with EN4.
In addition, salinity is roughly about 0.2 psu higher in the GIN Seas in $HR_{ide}$ (also in $HR_{tke}$ and $HR_{kpp}$). These slightly colder
but more saline water masses provide more favourable conditions for deep convection in the GIN Seas.

All three sensitivity simulations further show enhanced flow in the western part of the subpolar gyre, especially in the
Labrador Sea (Fig. 12b-d), where they also simulate deeper mixed layers. The flow weakens, however, in the eastern part of
the subpolar gyre, most pronounced in $HR_{ide}$.

The sensitivity simulations produce a stronger and deeper reaching upper branch of the AMOC of about $> 18\,Sv$ at $26.5\,°N$
(Fig. 13) compared with about $> 15\,Sv$ in $HR_{pp}$. A stronger upper cell may imply a stronger northward heat transport, whereas
a deeper upper cell indicates a stronger southward transport of NADW. We note, however, that the bottom cell is weaker in all
sensitivity simulations, which is probably due to a stronger vertical mixing with NADW, causing the Antarctic Bottom Water
(AABW) to vanish. Possibly, simulation length of 100 years are too short to see pronounced effects in the deep ocean. Over
longer simulation periods (several centuries), however, the additional mixing from internal waves might affect the diapycnal
diffusion of the upwelling deep water, e.g. in the Pacific.

Although IDEMIX affects the water mass properties and deep convection in the SPNA, there is hardly any difference to
$HR_{tke}$ and $HR_{kpp}$. This result challenges the relationship described by Kim et al. (2015), who link reduced vertical diffusivity
under sea ice in the Arctic Ocean to a doming of the isopycnals in the Nordic Seas, thus leading to higher overflow volumes
and a 14 % stronger upper cell of the AMOC.

### 4.2.2   Overflows from the Nordic Seas

A substantial contribution to the NADW constitutes the Denmark Strait overflow water (DSOW; $\sigma > 1027.8\,kg\,m^{-3}$), which
accounts for about half of the overflows from the Nordic Seas (Hansen et al., 2004), being its densest water mass. The other
major overflow pathway across the Greenland-Iceland-Scotland Ridge is the through the Faroe Shetland Channel (FSC; $\sigma >$





$1027.75\,\mathrm{kg\,m^{-3}}$) and through the Faroe Bank Channel (FBC; $\sigma > 1027.75\,\mathrm{kg\,m^{-3}}$). The enhanced MLDs in the Nordic Seas suggests stronger deep convection in the Nordic Seas in $\mathrm{HR_{kpp}}$, $\mathrm{HR_{tke}}$, and $\mathrm{HR_{ide}}$ and hence higher overflow volumes. We

applied Welch's two-sided t-tests with $\alpha = 0.05$ ($n = 20$) to test for significant differences in the simulated overflows. See Tab. A1 for all test results.

First we note that all simulations underestimate the observed DSOW volume transport (about $3.2$ to $3.4\,\mathrm{Sv}$) by about $1\,\mathrm{Sv}$ (see Tab. 2). Compared with $\mathrm{HR_{pp}}$, however, we find about 10 to 20 % higher DSOW transports in $\mathrm{HR_{kpp}}$ ($p < 0.01$), $\mathrm{HR_{ide}}$ ($p < 0.01$) and especially in $\mathrm{HR_{tke}}$ ($p < 0.01$). The transports in the sensitivity simulations themselves, however, differ not

significantly ($p$-values of 0.13 to 0.52). The higher amount of DSOW in the sensitivity simulations might thus explain at least partly the stronger upper cell of the AMOC, and in particular the AMOC strength around $60\,^\circ\mathrm{N}$.

Despite being overestimated compared to observations, the FSC overflows are of about similar magnitude in the simulations ($3.2$ to $3.3\,\mathrm{Sv}$), except for $\mathrm{HR_{tke}}$, which produces about 10 % higher ($3.5\,\mathrm{Sv}$) overflow transport ($p < 0.01$). The FBC overflows are about 15 to 20 % lower in the models ($1.7$ to $1.9\,\mathrm{Sv}$) than the observed $2.2\,\mathrm{Sv}$ by Hansen et al. (2016). The deviations

between the models are of the order of 10 %, with a higher mean transport in $\mathrm{HR_{tke}}$ ($p < 0.01$) and a lower transport in $\mathrm{HR_{ide}}$ ($p < 0.01$).

Overall, these results suggest about 10 % higher overflow transports across the Greenland-Iceland-Scotland ridge in the sensitivity simulations, which contribute to a stronger upper cell of the AMOC.

### 4.3 Southern Ocean and Weddell Sea

#### 4.3.1 Weddell Sea

A long lasting problem in ocean modelling is a too frequent semi-permanent Weddell Sea polynya caused by a spurious deep convection bringing warm Circumpolar Deep Water close to the surface (Sallée et al., 2013; Kjellsson et al., 2015; Heuzé et al., 2015; Stössel et al., 2015; Naughten et al., 2018). Possible explanations are insufficient freshwater supply (Kjellsson et al., 2015), mainly due to a lack of glacier melt water (Stössel et al., 2015), and insufficient wind mixing in summer (Timmermann

and Beckmann, 2004; Sallée et al., 2013; Kjellsson et al., 2015), which causes a high salinity bias in the mixed layer that erodes the stratification; see a more detailed discussion in Gutjahr et al. (2019). In contrast to this view, Dufour et al. (2017) argue that deep convection in the Weddell Sea does not necessarily lead to an open-ocean polynya, because strong vertical mixing in low-resolution models inhibits the built-up of a subsurface heat reservoir that would be necessary for intermittent Weddell Sea polynyas.

We do not expect realistic representation of the Weddell Sea polynya in our MPI-ESM1.2-HR 1950s control simulations, because they should develop intermittently only under pre-industrial conditions and grow out from Maud Rise polynyas (de Lavergne et al., 2014; Gordon, 2014; Kurtakoti et al., 2018; Campbell et al., 2019; Cheon and Gordon, 2019; Jena et al., 2019), for which high resolution ($0.1\,^\circ$) is required (Stössel et al., 2015; Dufour et al., 2017).

Although all simulations produce these semi-permanent Weddell Sea polynyas and thus too deep mixed layers (Fig. 14), the

area of excessive deep-convection is reduced in $\mathrm{HR_{ide}}$ (Fig. 14e). Likewise, too deep mixed layers are simulated in the Ross





Sea, except shallower mixed layers in $HR_{tke}$; without further reduction when using IDEMIX ($HR_{ide}$). The Weddell Gyre is also linked to the Antarctic Circumpolar Current (ACC; Orsi et al., 1993; Cheon et al., 2019), because it controls the inflow of relatively warm and saline Circumpolar Deep Water into the inner Weddell Sea, possibly eroding the weak stratification and triggering deep convection. The simulated ACC transports through Drake passage (Tab. 3) are close to the recently observed

$173.31 \pm 10.7\,\mathrm{Sv}$ (Donohue et al., 2016), whereby $HR_{tke}$ achieves the best estimate with $174\,\mathrm{Sv}$. Models with lower convection in the Weddell Sea produce lower transports of about $161 - 163\,\mathrm{Sv}$ ($HR_{pp}$ and $HR_{ide}$) and the model with stronger convection ($HR_{kpp}$) produces a much higher transport of $192\,\mathrm{Sv}$.

One possible explanation why $HR_{ide}$ simulates less convection in the Weddell Sea is that IDEMIX creates a stronger mixing above the shelf, whereby near-surface freshwater spreads much more efficiently in the lateral direction towards the centre of

the Weddell Gyre (not shown). This higher amount of freshwater in the Weddell Gyre increases the formation of sea ice, which then isolates the ocean from further heat loss and thus impedes convection. In $HR_{ide}$, the average sea ice concentration in September is about 50 to $70\,\%$ in the Weddell Gyre (Fig. 15), whereas it is considerably lower in the other simulations with concentrations of 20 to $50\,\%$. Although the sea ice concentration is still too low, IDEMIX improves the near-surface freshwater supply.

### 4.3.2 Deep Mixing Band

Another challenge for current ocean models is the representation of the Deep Mixing Band (DMB; DuVivier et al., 2018), which extends from the western Indian Ocean to the eastern Pacific Ocean and reaches MLDs of more than $700\,\mathrm{m}$ (Holte et al., 2017, Fig. 14a). The DMB builds up over the winter months and is deepest in September. Subantarctic Mode Water (SAMW; McCartney, 1977) forms in the DMB near the Subantarctic Front, just north of the ACC. The SAMW acts as an important

carbon sink (e.g. Sabine et al., 2004) and it ventilates the mid-deep ocean (e.g. Sloyan and Rintoul, 2001; Jones et al., 2016), replenishing oxygen and nutrients (e.g. Sarmiento et al., 2004). Although it was shown that high resolution ($0.1°$) leads to deeper and thus more realistic MLDs in the DMB (Li and Lee, 2017; DuVivier et al., 2018; Gutjahr et al., 2019), it is thought that fundamental vertical physics are missing in ocean models (DuVivier et al., 2018).

Although $HR_{pp}$ reproduces the mixing band in the Indian Ocean, the mixed layers are too shallow almost entirely in the

Pacific Ocean sector (Fig. 14b). In contrast, the sensitivity simulations improve the representation of the DMB in the Pacific Ocean and particularly in the Indian Ocean. The MLDs are close to observations (Fig. 14c-e), albeit with a too wide DMB, which is probably caused by insufficient model resolution, since it becomes much narrower when an eddy-resolving model is used (Gutjahr et al., 2019). The choice of a mixing scheme other than PP has little influence on the MLDs, except south of Tasmania, where TKE appears to produce the deepest mixing layers.

Although not observed by Argo floats, but common to all simulations are deeper mixed layers north of $50°$S in the Pacific Ocean east of New Zealand and in the South Atlantic. However, it should be kept in mind that comparing model simulations with Argo float data is always difficult, because the floats do not measure continuously in time and space. And furthermore, a difference could arise due to the different time periods we compare.





## 5 Conclusions

We have compared the effect of four different ocean vertical mixing schemes (PP, KPP, TKE, TKE+IDEMIX) on the ocean mean state in MPI-ESM1.2-HR simulations. The ocean mixing library CVMix (Griffies et al., 2015), which we extended with the TKE and TKE+IDEMIX schemes, allowed for a side-by-side comparison of these schemes.

Although we conclude that shortcomings with respect to model resolution determine the global bias pattern in the ocean in our coupled simulations, the choice of the ocean vertical mixing scheme plays an important role for regional biases. In general,

we find little sensitivity of the ocean surface, but considerable effects for the interior ocean. Overall, we cannot classify any scheme as superior to the others, as these schemes modify biases that vary with region or variable but produce a similar global-scale bias pattern.

All simulations show strong temperature and salinity biases in the Atlantic that originate from insufficient representations of the Mediterranean overflow water (MOW) and Agulhas leakage. The biases of MOW can be traced from the Atlantic Ocean

to the Arctic Ocean via the GIN Seas and the Nordic Seas. Based on a recent study with an eddy-resolving version of MPI-ESM1.2 (Gutjahr et al., 2019), we know that most of these biases diminish at such a resolution. However, eddy-resolving resolution is computationally demanding and therefore it is important to investigate whether improved parameterisation can improve some of the hydrographic errors.

The sensitivity simulations using KPP, TKE, and TKE+IDEMIX, produce deeper than observed MLDs in the subpolar North

Atlantic and the Nordic Seas, causing about 10 % higher overflow volumes and thus a stronger upper cell of the AMOC. In the Southern Ocean, however, the MLDS in the Deep Mixing Band agree better with observations from Argo floats.

Although extending the TKE scheme with IDEMIX does not improve these biases on a global scale, we notice some regional improvements. Most pronounced, IDEMIX produces a more heterogeneous spatial pattern of vertical diffusivity, with generally lower values in deep and flat basins and elevated turbulence above rough topography. This spatial pattern is particularly

evident for the Arctic Ocean, which is more consistent with micromeasurements, without lowering the background artificially. Furthermore, IDEMIX improves the circulation and mixing in the Nordic Seas and in the Fram Strait, which reduces the warm bias of the Atlantic water layer in the Arctic Ocean. This reduction of biases is similar to what was recently shown with high-resolution, eddy-resolving ocean models (Wang et al., 2018; Gutjahr et al., 2019). However, the main advantages of IDEMIX is its energetically consistent link to TKE and its flexibility that allows additional energy inputs, e.g. from the mesoscale eddy

field.

*Code and data availability.* Model codes developed at MPI-M are intellectual property of MPI-M. Permission to access the MPI-ESM source code can be requested after registering at the MPI user forum (https://www.mpimet.mpg.de/en/science/models/mpi-esm/users-forum/, last accessed August 2020) and may be granted after accepting the MPI-M Software License Agreement (https://www.mpimet.mpg.de/fileadmin/models/MPIESM/mpi-m_sla_201202.pdf).

After registration, the model source code is accessible from https://code.mpimet.mpg.de/documents/521 using branch *mpiesm-1.2.01-cvmix_GMD* for the KPP, TKE, and TKE+IDEMIX simulations and *mpiesm-1.2.01-primavera_PP_GMD* for the PP simulation. Primary





data and scripts used in the analysis, and other supplementary information that may be useful in reproducing the author's work can be obtained from MPI PuRe (http://hdl.handle.net/21.11116/0000-0006-DB1E-3). Initial fields and all forcing fields can be downloaded from DKRZ LTA DOCU (https://cera-www.dkrz.de/WDCC/ui/cerasearch/entry?acronym=DKRZ_LTA_944_ds00001). The model code used for the simulations in the paper and the primary data have been provided to the anonymous reviewer and the topical editor.

## Appendix A: Ocean vertical mixing schemes in MPI-ESM1.2

In this section we give a brief summary of the ocean vertical schemes that we have implemented into MPI-ESM1.2. The vertical mixing schemes were implemented via the Community Vertical Mixing (CVMix) library (Griffies et al., 2015; Van Roekel et al., 2018), except for the PP scheme. The KPP schema was already part of the CVMix library, which we therefore extended with the TKE and IDEMIX scheme. This extension will become part of the official CVMix library.

The schemes below aim to parameterise the vertical turbulent fluxes following this general flux-gradient or K-profile approach:

$$\overline{w'\lambda'} = -K_\lambda \left( \frac{\partial \Lambda}{\partial z} \right) + \Gamma_\lambda, \tag{A1}$$

with $w'$ the vertical turbulent velocity, $\lambda'$ the turbulent fluctuation of a quantity, $\lambda$ a grid-scale quantity and the turbulent exchange coefficient ($K_\lambda > 0$) or also termed eddy viscosity for momentum flux and eddy diffusivity for tracer fluxes, such as temperature or salinity. $\Gamma_\lambda$ represents any flux not proportional to the local gradient $\partial_z \Lambda$ and is referred to as *nonlocal flux*. In our comparison, $\Gamma_\lambda$ is only accounted for in the KPP scheme (A2).

### A1 Pacanowski and Philander (1981, PP) scheme

In our control simulation the vertical turbulent diffusion and viscosity are based on a modified version of the Richardson-number dependent formulation by Pacanowski and Philander (1981) (PP scheme). The modifications are that (1) the vertical diffusivity is not dependent on the vertical viscosity, and (2) that the turbulent mixing in the ocean mixed layer is assumed to depend on the cube of the 10 m wind speed (Marsland et al., 2003). This dependency decays exponentially with depth and with potential density difference to the surface. Since the classical approach by Pacanowski and Philander (1981) underestimates the turbulent mixing close to the surface (Marsland et al., 2003), this additional wind induced mixing ($\kappa_w$) is added, and defined as:

$$K_w(1) = (1 - A)^2 w_t U_{10}^3 \tag{A2}$$

$$K_w(k) = K_w(k-1) \frac{\frac{\lambda}{\Delta z}}{\frac{\lambda}{\Delta z} + \delta_z \rho} e^{\frac{\lambda}{z_0}} \tag{A3}$$

with $k = 2, 3, \ldots, n$ the vertical model level, $\Delta z$ the layer thickness, $A$ the fractional sea ice concentration, $U_{10}$ the 10 m wind speed, $w_t = 0.5 \cdot 10^{-3}/6.0^3$, $\lambda = 0.03$, and $z_0 = 40.0$ (e-folding depth) are adjustable parameters, and $\delta_z \rho$ the local static stability.



The total equation for the eddy vertical diffusivity then reads:

$$K_d(z) = \frac{d_{v,0}}{(1 + c_d Ri(z)^2)} + K_w(z) + d_{v,b}, \tag{A4}$$

with $d_{v,0} = 0.2 \cdot 10^{-2}\,\mathrm{m}^2\,\mathrm{s}^{-1}$, $c_d = 5.0$, and the background diffusivity $d_{v,b} = 5 \cdot 10^{-5}\,\mathrm{m}^2\,\mathrm{s}^{-1}$. The eddy vertical viscosity is parameterised as:

$$K_v(z) = \frac{a_{v,0}}{(1 + c_a Ri(z)^3)} + K_w(z) + a_{v,b}, \tag{A5}$$

with $a_{v,0} = 0.2 \cdot 10^{-2}\,\mathrm{m}^2\,\mathrm{s}^{-1}$, $c_a = 5.0$, and $a_{v,b} = 1.05 \cdot 10^{-5}\,\mathrm{m}^2\,\mathrm{s}^{-1}$ the background viscosity. The eddy coefficients $K_d$ and $K_v$ are partially relaxed to the value at the previous time step by use of a time filter to avoid $2\Delta t$ oscillations (Marsland et al., 2003). Convection is parameterised as enhanced diffusivity ($K_d = 0.1\,\mathrm{m}^2\,\mathrm{s}^{-1}$) for the PP scheme.

## A2  K-profile parameterisation (KPP) scheme

The second simulation uses the K-profile parameterisation (KPP scheme) from Large et al. (1994) for the mixed layer. In general, a turbulent flux ($\overline{w'\lambda'} = \overline{w\lambda}$) of a quantity $\Lambda$ (momentum, scalar tracers) are parameterised as:

$$\overline{w\lambda} = -K_\lambda \left( \frac{\partial \Lambda}{\partial z} \right) + K_\lambda^{\text{non-local}} \gamma_\lambda, \tag{A6}$$

with $K_\lambda$ the local diffusivity for tracers or viscosity for momentum, $z$ the depth, $K_\lambda^{\text{non-local}}$ the non-local diffusivity/viscosity, and a non-local term $\gamma_\lambda$. While the local flux (first term on the right hand side) depends directly on the local gradient of a quantity, the non-local flux (second term on the right hand side) redistributes the surface fluxes throughout the whole surface boundary layer, for example due to convection (see below).

The local diffusivity ($K_\lambda$) is calculated as:

$$K_\lambda(\sigma) = h\, w_\lambda(\sigma) G(\sigma), \tag{A7}$$

with $\sigma = z/h$ the dimensionless vertical coordinate ($0 \leq \sigma \leq 1$), $z$ the depth below the sea surface, $h$ the ocean boundary layer depth, $w_\lambda$ a vertical turbulent velocity scale (either for scalar tracers or momentum), and $G(\sigma) = \sigma(1-\sigma)^2$ a universal shape function. Oftentimes, and also in our implementation, it is assumed that $K_\lambda^{\text{non-local}} = K_\lambda$ (Griffies et al., 2015).

The ocean boundary layer depth $h$ is determined at the depth $z$ where the bulk Richardson-number $Ri_b$ becomes larger than a critical Richardson-number $Ri_c = 0.7$. The bulk Richardson-number at depth $z$ is defined as:

$$Ri_b(z) = \frac{(z - z_{sl})\,[B_{sl} - B(z)]}{|\mathbf{U}_{sl} - \mathbf{U}(z)|^2 + U_t(z)^2}, \tag{A8}$$

with $z_{sl}$ the depth at the centre of the surface layer (defined as $0 \leq z \leq \epsilon h$), where we assume that the surface layer is 10 % ($\epsilon = 0.1$) of the ocean boundary layer depth $h$, as in Large et al. (1994). Since the calculation of the ocean boundary layer





depth $h$ requires $Ri_b$, which itself requires $h$, we face a cyclic problem. To solve this problem, we follow the column sampling method recommended by Griffies et al. (2015) (see details in their description).

$B_{sl}$ is the surface layer averaged buoyancy flux, $B(z)$ the local buoyancy flux, $U_{sl}$ the surface layer averaged velocity, $U(z)$

the local velocity and $U_t(z)$ a parameterisation for unresolved turbulent vertical velocity shear that reduces the bulk Richardson number (see e.g. Griffies et al. (2015) for the definition of the unresolved turbulent shear).

The vertical turbulent velocity scale $w_\lambda$ is calculated as follows:

$$w_\lambda(\sigma) = \frac{\kappa u_*(1-A)^2}{\Phi_\Lambda(\zeta)},$$  (A9)

with $\kappa = 0.4$ the von-Kármán constant, $u_*$ the surface friction velocity that reduces with increasing fractional sea ice ($A$), $\Phi_\Lambda$

a dimensionless similarity function (momentum or scalar tracer), depending on $\zeta = \sigma h/L$, with $L = \frac{u_*^3}{\kappa B_f}$ the Monin-Obukhov length scale. $B_f$ is the buoyancy forcing in the ocean boundary layer. Under neutral forcing ($\zeta = 0 \to \Phi_\Lambda(0) = 1$), Eq. A9 reduces to $w_\lambda(\sigma) = \kappa u_*(1-A)^2$. We use the similarity functions of Large et al. (1994) (cf see Appendix B) for stable ($\zeta > 0$), unstable ($\zeta_m < \zeta < 0$ or $\zeta_s < \zeta < 0$), and very unstable conditions ($\zeta < 0$):

$$\Phi_m(\zeta) = \begin{cases} 1 + 5\zeta & \zeta > 0 \\ (1 - 16\zeta)^{-1/4} & \zeta_m < \zeta < 0 \\ (a_m - c_m\zeta)^{-1/3} & \zeta < -\zeta_m \end{cases}$$  (A10)

$$\Phi_s(\zeta) = \begin{cases} 1 + 5\zeta & \zeta > 0 \\ (1 - 16\zeta)^{-1/4} & \zeta_s < \zeta < 0 \\ (a_s - c_s\zeta)^{-1/3} & \zeta < \zeta_s \end{cases}$$  (A11)

with $\zeta_m = -0.2$, $\zeta_s = -1.0$, $a_m = 1.26$, $c_m = 8.38$, $a_s = -28.86$, and $c_s = 98.96$. We do not match the diffusivities at the base of the mixed layer to avoid overshooting tracers, as recommended by Griffies et al. (2015).

For the non-local flux term in Eq. A6, it is assumed that $K_\lambda = K_\lambda^{\text{non-local}}$, so that this term simplifies to: $K_\lambda \gamma_\lambda$. The non-local flux $\gamma_\lambda$ is only non-zero if $B_f < 0$ (buoyancy gain at the surface) and only for scalar tracers such as temperature, $\theta$, or salinity,

$s$; for momentum it is set to zero. With the assumption $K_\lambda = K_\theta = K_s$ and a universal shape function ($G(\sigma)$), the non-local fluxes take the form:

$$\overline{w\theta}^{\text{non-local}} = K_\lambda \gamma_\theta = -G(\sigma)C_s\left(\frac{Q^{\text{heat}}}{\rho_0 C_P^0}\right)$$  (A12)

$$\overline{ws}^{\text{non-local}} = K_\lambda \gamma_s = -G(\sigma)C_s Q^s,$$  (A13)

with the non-local terms for temperature ($\gamma_\theta$) and salinity ($\gamma_s$), a dimensionless coefficient $C_s = C_*\kappa(c_s\kappa\epsilon)^{1/3}$ with a di-

mensionless constant $C_* = 10$, $\rho_0 = 1025\,\text{kg}\,\text{m}^{-3}$ the reference sea water density, $C_P^0$ the seawater heat capacity at constant



pressure ($\mathrm{J\,kg^{-1}\,{}^\circ C^{-1}}$), $Q_s$ the mass flux of salt ($\mathrm{kg\,m^{-2}s^{-1}}$) and the heat flux $Q^{\mathrm{heat}}$ ($\mathrm{W\,m^{-2}}$) that considers penetrative shortwave radiation. See further details on the KPP scheme in Griffies et al. (2015) and Van Roekel et al. (2018).

Below the mixed layer, we use the PP scheme with the enhanced diffusivity parameterisation for convection, as described in section A1.

### A3  TKE and IDEMIX schemes

A schematic overview of the TKE and IDEMIX schemes are depicted in Fig. A1. IDEMIX parameterises the internal wave energy ($E_{iw}$) in terms of a budget equation:

$$\frac{\partial E_{iw}}{\partial t} = \nabla_h \nu_0 \tau_h \nabla_h \nu_0 E_{iw} + \frac{\partial}{\partial z}\left(c_0 \tau_v \frac{\partial c_0 E_{iw}}{\partial z}\right) - \epsilon_{iw} \tag{A14}$$

with $\nu_0$ the lateral group velocity, $\tau_h$ a lateral time scale on which lateral anisotropies are eliminated by nonlinear wave-wave interactions, $c_0$ the weighted average vertical group velocity, $z$ the vertical coordinate, $\tau_\nu$ a time scale of on the order of days, and $\epsilon_{iw} = -\alpha E_{iw}^2$ the dissipation of internal wave energy, with $\alpha$ a structure function depending on the stratification (for details see (Olbers and Eden, 2013)). The first and second term on the right-hand side (RHS) parameterise the horizontal and vertical propagation of internal waves, respectively. Internal wave energy is dissipated by the last term on the rhs of Eq. A14. This term acts as an energy transfer from internal wave energy to turbulent kinetic energy (see Eq. A15 below).

Internal waves are forced in IDEMIX by surface and bottom fluxes applied as boundary conditions in the second term on the RHS of Eq. A14. Currently, we use time constant fields for the energy fluxes at the surface and at the bottom, as in Olbers and Eden (2013). To this end, the energy flux which we use as upper boundary condition is 20 % of the wind input into the inertial band of the mixed layer (Jochum et al., 2013), as determined by Rimac et al. (2013) (inertial pumping mechanism). We neglect, however, other sources exciting internal waves near the surface, for instance buoyancy plumes that overshoot the mixed layer base, vertical roll vortices of turbulent eddies, or Langmuir circulation that undulates internal waves (Czeschel and Eden, 2019). At the bottom, barotropic tides interact with the bottom roughness and convert energy to internal waves. This energy flux is prescribed from Jayne (2009), with the constraint that this energy source is not flow-aware.

In commonly used vertical mixing schemes, such as KPP (Large et al., 1994) or TKE (Gaspar et al., 1990), the breaking of internal waves is usually parameterised by simply assuming a constant background diffusivity (either a scalar or profile) or constant background TKE value. By using TKE with IDEMIX, the constant background diffusivity is replaced by one that is diagnosed from the internal wave energy using the Osborn-Cox relation. We use the recommended parameter set from Pollmann et al. (2017), who compared TKE coupled to IDEMIX with Argo float data in standalone ocean simulations.

The modified TKE equation (Eden et al., 2014) then reads:

$$\frac{d\bar{E}_{tke}}{dt} = -\partial_z(\text{fluxes}) + c_u K (\partial_z \bar{\mathbf{u}}^2) - c_b K N^2 - \epsilon_{tke} + \epsilon_{iw}, \tag{A15}$$

with the dimensionless parameters $c_u$ and $c_b$, which are related by $c_u = c_b R_i / R_f$. The first term on the RHS describes the redistribution of TKE in the vertical. Surface fluxes enter as boundary conditions to this term. The second term describes the





vertical momentum fluxes acting on the shear of the mean flow, transferring energy from the mean flow to TKE. The third term is the buoyancy term that transfers energy to the potential energy of the mean flow, thereby decreasing TKE. The dissipation of TKE (fourth term) is parameterised as $\epsilon_{tke} = \bar{E}_{tke}^{2/3} L^{-1}$ with the mixing length $L = \sqrt{2\bar{E}_{tke}/N^2}$ (Blanke and Delecluse, 1993; Eden et al., 2014). The last term on the RHS is then the new contribution from IDEMIX, which is absent when using TKE without IDEMIX.

The diffusivity is parameterised as $K = \bar{E}_{tke}^{1/2} L$ by assuming the same mixing length as for the dissipation. If TKE is used alone without being combined with IDEMIX, then a background diffusivity is assumed to represent internal wave breaking (Eden et al., 2014). When TKE is used alone without being coupled to IDEMIX, the turbulent kinetic energy is set to a background value of $E_{min} = 10^{-6}\,\mathrm{m^2 s^{-2}}$. The corresponding diffusivity in this case reads $K = \sqrt{2}c_b E_{min}/N$.

*Author contributions.* JJ, JvS and OG designed the experiments. DP, KL, and OG performed the simulations. OG, NB and HH have implemented the KPP, TKE, and IDEMIX mixing parameterisations in MPIOM. OG prepared the manuscript with contributions from all co-authors.

*Competing interests.* The authors declare that they have no conflict of interest.

*Acknowledgements.* We thank the German Computing Centre (DKRZ) for providing the computing resources. This research was funded by the EU Horizon 2020 project PRIMAVERA (grant number 641727). This paper is a contribution to the project S2 (Improved parameterisations and numerics in climate models) of the Collaborative Research Centre TRR 181 "Energy Transfer in Atmosphere and Ocean" funded by the Deutsche Forschungsgemeinschaft (DFG, German Research Foundation) - project number 274762653. JJ, OG, and KL further acknowledge support by the German BMBF RACE-II project (FKZ 03F0729D). We thank Jöran März for the internal review of the manuscript and constructive comments before submission. The Argo float data were made freely available by the International Argo Program and the national programs that contribute to it (http://www.argo.ucsd.edu, http://argo.jcommops.org)



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





**Table 1.** Overview of the 100 year long control simulations conducted with MPI-ESM1.2-HR. All simulations use a T127 (about 100 km) resolution in the atmosphere and a resolution of $0.4\,^\circ$ (about 40 km) on a tripolar grid (TP04) in the ocean. The number of vertical levels are 95 in the atmosphere and 40 in the ocean, respectively. All models were analysed for the model years 80–100.

| Experiment | Ocean mixing scheme | Description | Reference |
|---|---|---|---|
| $HR_{pp}$ | PP | reference simulation | (Pacanowski and Philander, 1981) |
| $HR_{kpp}$ | KPP | uses PP scheme below mixed layer | (Large et al., 1994; Griffies et al., 2015) |
| $HR_{tke}$ | TKE | background diffusivity $K = \sqrt{2}E_{min}/N$ with $E_{min} = 10^{-6}\,\mathrm{m^2s^{-2}}$ | (Gaspar et al., 1990) |
| $HR_{ide}$ | TKE + IDEMIX | $E_{min} = 0\,\mathrm{m^2s^{-2}}$; prognostic simulation of internal gravity waves | (Eden et al., 2014) |





**Table 2.** Time-averaged volume transport (1 Sv$= 10^6 \, \mathrm{m^3 \, s^{-1}}$) of Denmark Strait Overflow Water (DSOW), Faroe Bank Channel (FBC) and Faroe-Shetland Channel (FSC) overflow from simulations with MPI-ESM1.2-HR. Hansen et al. (2017) note that measurements by Rossby and Flagg (2012) and Childers et al. (2014) include a closed circulation on the Faroe Shelf (0.6 Sv) and flow on the Scottish Shelf, which are not included in measurements by Berx et al. (2013) and Hansen et al. (2015). A standard deviation based on annual averages is given for the simulations.

| Observations/Experiment | DSOW | FBC | FSC |
| --- | --- | --- | --- |
| Jochumsen et al. (2017) | $-3.2 \pm 0.5$ | - | - |
| Jochumsen et al. (2012) | $-3.4$ | - | - |
| Hansen et al. (2016) | - | 2.2 | - |
| Berx et al. (2013); Hansen et al. (2015) | - | - | 2.7 |
| Rossby and Flagg (2012) | - | - | 0.9 |
| Childers et al. (2014) | - | - | 1.5 |
| HR$_{pp}$ | $-1.8 \pm 0.2$ | $1.8 \pm 0.1$ | $3.2 \pm 0.4$ |
| HR$_{kpp}$ | $-2.1 \pm 0.4$ | $1.8 \pm 0.2$ | $3.3 \pm 0.3$ |
| HR$_{tke}$ | $-2.2 \pm 0.3$ | $1.9 \pm 0.1$ | $3.5 \pm 0.2$ |
| HR$_{ide}$ | $-2.1 \pm 0.3$ | $1.7 \pm 0.1$ | $3.3 \pm 0.3$ |





**Table 3.** Time-averaged volume transport ($1\,\mathrm{Sv} = 10^6\,\mathrm{m^3\,s^{-1}}$) of the Antarctic Circumpolar Current (ACC) through Drake Passage from observations and simulations with MPI-ESM1.2-HR.

| Experiment | Mean | Standard deviation |
|---|---|---|
| Donohue et al. (2016) | 173.3±10.7 | - |
| HR$_{pp}$ | 161.41 | 2.14 |
| HR$_{kpp}$ | 191.97 | 2.99 |
| HR$_{tke}$ | 174.31 | 2.63 |
| HR$_{ide}$ | 163.54 | 4.39 |





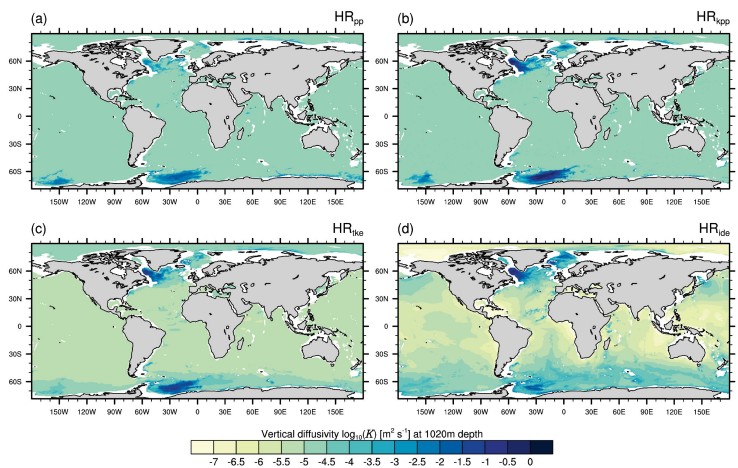

**Figure 1.** Time-averaged vertical diffusivity $\log_{10}(K)$ (m$^2$ s$^{-1}$) at a depth of 1020 m in the MPI-ESM1.2-HR simulations for (a) HR$_{pp}$, (b) HR$_{kpp}$, (c) HR$_{tke}$, and (d) HR$_{ide}$.





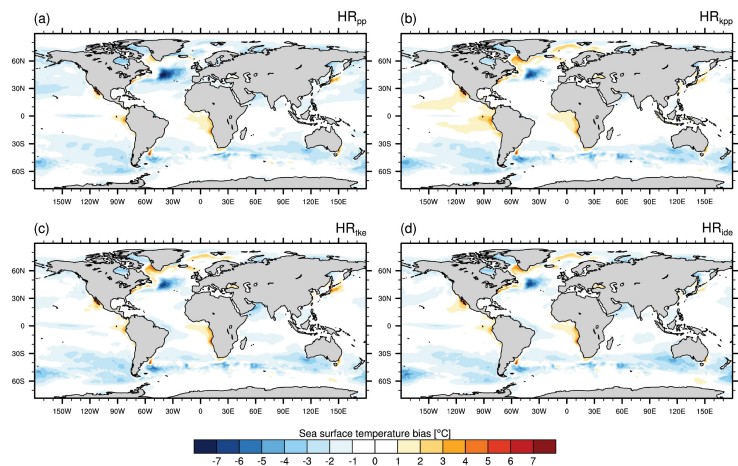

**Figure 2.** Time-averaged sea surface temperature bias of MPI-ESM1.2-HR minus EN4 (1945-1955) for (a) HR$_{pp}$, (b) HR$_{kpp}$, (c) HR$_{tke}$, and (d) HR$_{ide}$.



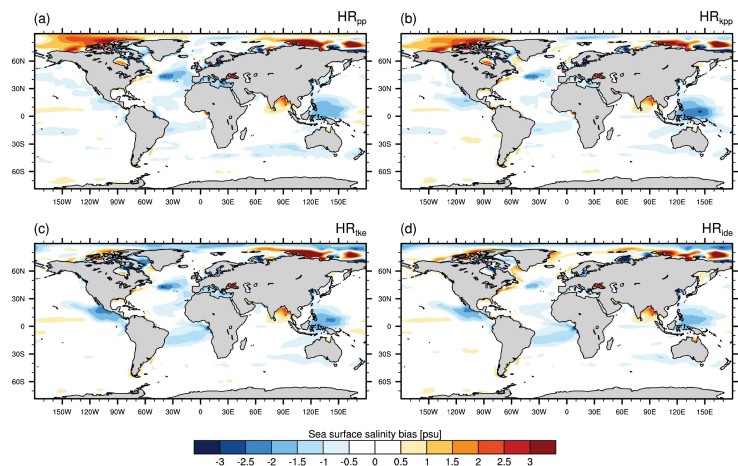

**Figure 3.** Time-averaged sea surface salinity bias of MPI-ESM1.2-HR minus EN4 (1945-1955) for (a) HR$_{pp}$, (b) HR$_{kpp}$, (c) HR$_{tke}$, and (d) HR$_{ide}$.



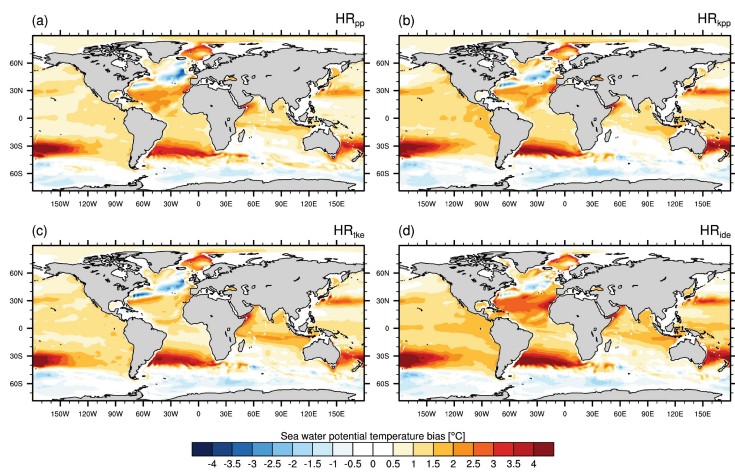

**Figure 4.** Time-averaged potential temperature bias of MPI-ESM1.2-HR minus EN4 (1945-1955) at a depth of 740 m for (a) $HR_{pp}$, (b) $HR_{kpp}$, (c) $HR_{tke}$, and (d) $HR_{ide}$.





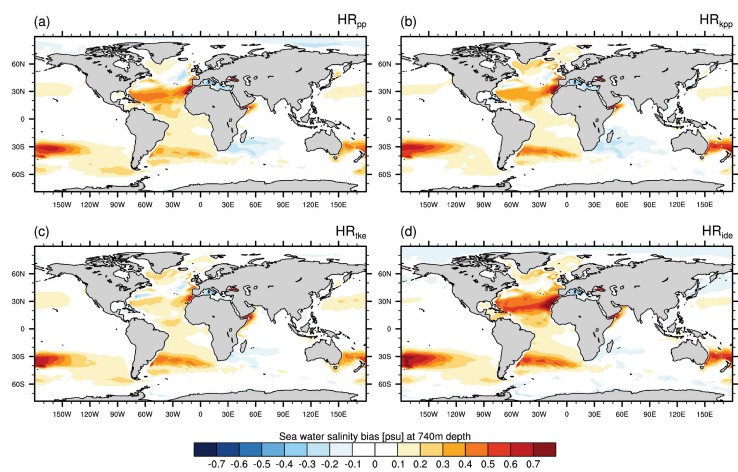

**Figure 5.** Time-averaged salinity bias of MPI-ESM1.2-HR minus EN4 (1945-1955) at a depth of 740 m for (a) HR$_{pp}$, (b) HR$_{kpp}$, (c) HR$_{tke}$, and (d) HR$_{ide}$.



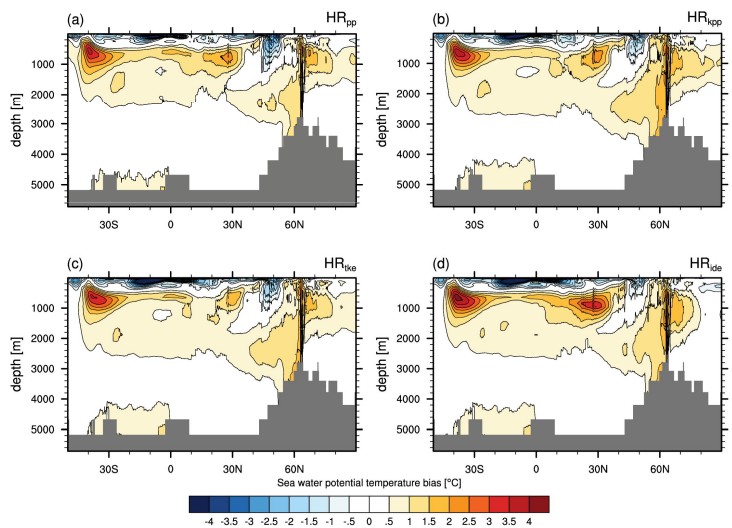

**Figure 6.** Zonal-mean potential temperature bias of MPI-ESM1.2-HR minus EN4 (1945-1955) in the Atlantic and Arctic Ocean for (a) HR$_{pp}$, (b) HR$_{kpp}$, (c) HR$_{tke}$, and (d) HR$_{ide}$.

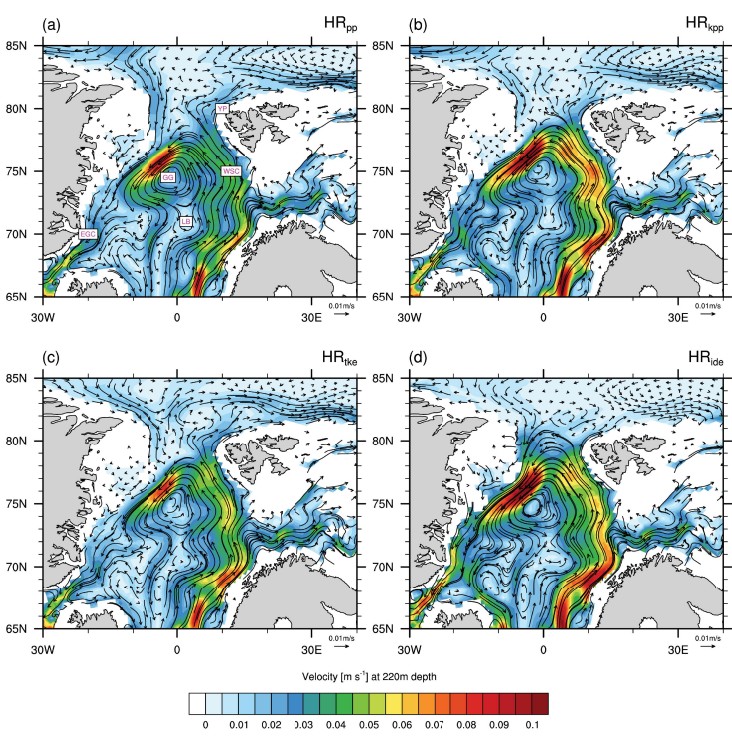

**Figure 7.** Time-averaged circulation in the Nordic Seas and at Fram Strait at a depth of 220 m simulated by (a) $HR_{pp}$, (b) $HR_{kpp}$, (c) $HR_{tke}$, and
(d) $HR_{ide}$. WSC: West Spitsbergen Current, EGC: East Greenland Current, GG: Greenland Gyre, LB: Lofoten Basin, YP: Yermak Plateau.

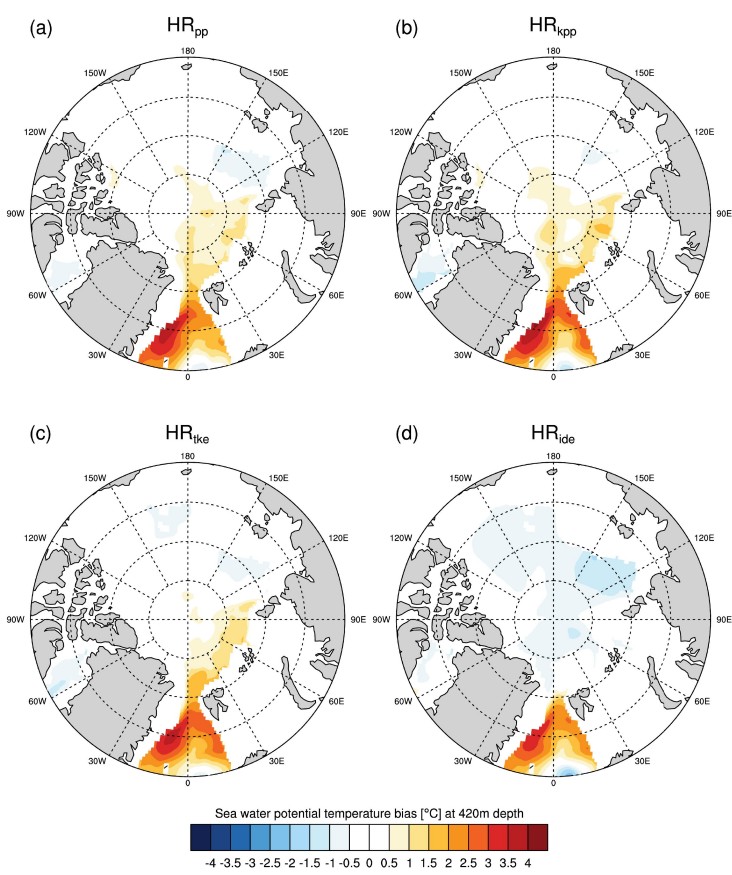

**Figure 8.** Time-averaged potential temperature bias of MPI-ESM1.2-HR minus EN4 (1945-1955) at a depth of 420 m in the Arctic Ocean and at Fram Strait for (a) $HR_{pp}$, (b) $HR_{kpp}$, (c) $HR_{tke}$, and (d) $HR_{ide}$.



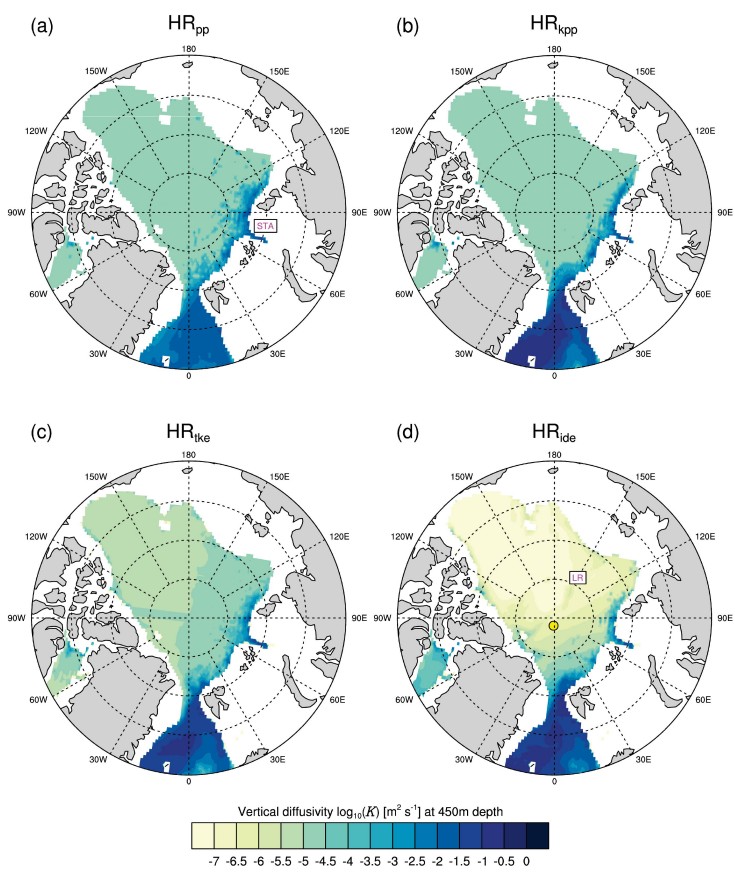

**Figure 9.** Time-averaged vertical diffusivity $\log_{10}(K)$ $(\mathrm{m^2\,s^{-1}})$ at a depth of $450\,\mathrm{m}$ in the Arctic Ocean and at Fram Strait simulated by MPI-ESM1.2 (a) HR$_{pp}$, (b) HR$_{kpp}$, (c) HR$_{tke}$, and (d) HR$_{ide}$. STA: St. Anna Trough, LR: Lomonosov Ridge. The yellow point marks the approximate position ($7\,^\circ$W, $89\,^\circ$N) of the Barneo ice camp in the Amundsen Basin, where $K$ on the $O(10^{-6}\,\mathrm{m^2\,s^{-1}})$ was measured below the mixed layer (Fer, 2009).



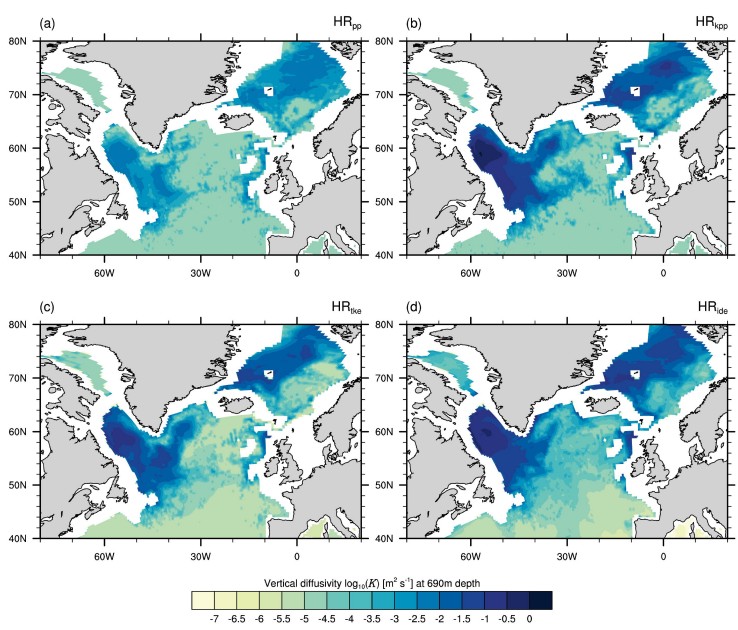

**Figure 10.** Time-averaged vertical diffusivity $\log_{10}(K)$ $(\mathrm{m^2\,s^{-1}})$ at a depth of $690\,\mathrm{m}$ in the subpolar North Atlantic simulated by MPI-ESM1.2 (a) $\mathrm{HR_{pp}}$, (b) $\mathrm{HR_{kpp}}$, (c) $\mathrm{HR_{tke}}$, and (d) $\mathrm{HR_{ide}}$.





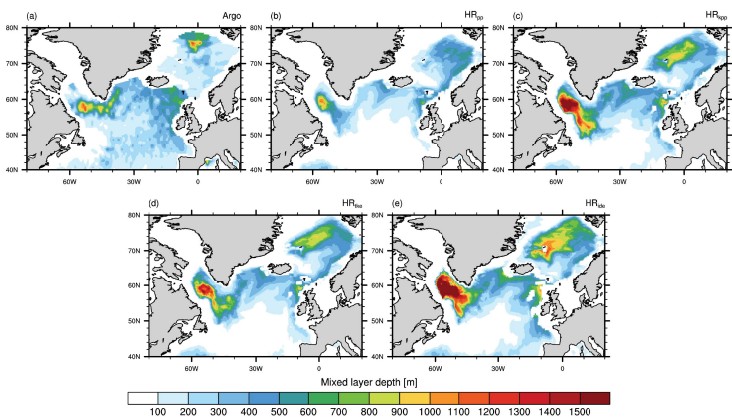

**Figure 11.** Time-averaged mixed layer depths (m) in March calculated (a) from $1° \times 1°$ Argo float data (Holte et al., 2017) (mean March from 2000 to 2018) by the density threshold method ($\sigma_t = 0.03 \, \mathrm{kg \, m^{-3}}$) and with $\sigma_t = 0.01 \, \mathrm{kg \, m^{-3}}$ from MPI-ESM1.2 (b) $HR_{pp}$, (c) $HR_{kpp}$, (d) $HR_{tke}$, and (e) $HR_{ide}$.

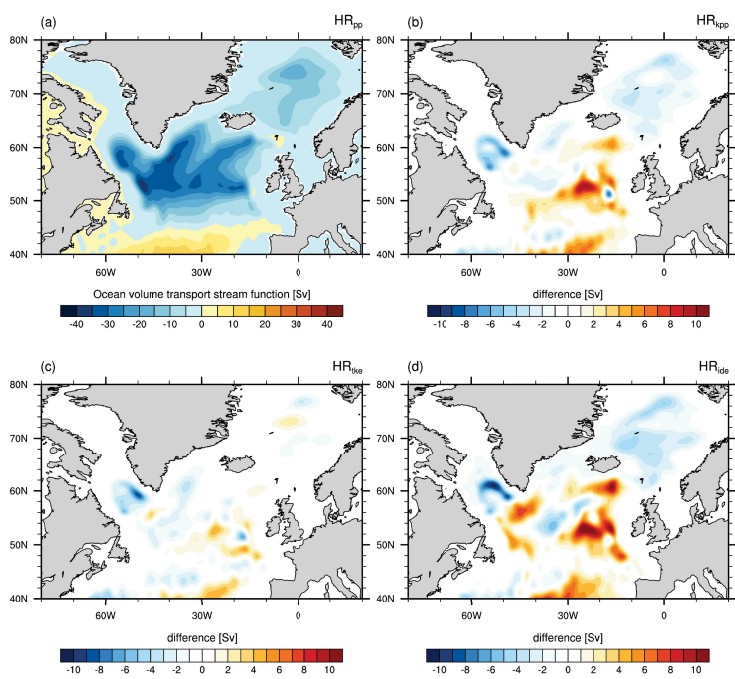

**Figure 12.** Time-averaged barotropic volume transport stream function (Sv) in the North Atlantic as simulated by MPI-ESM1.2 (a) $HR_{pp}$, and the differences "experiment - $HR_{pp}$" for (b) $HR_{kpp}$, (c) $HR_{tke}$, and (d) $HR_{ide}$.



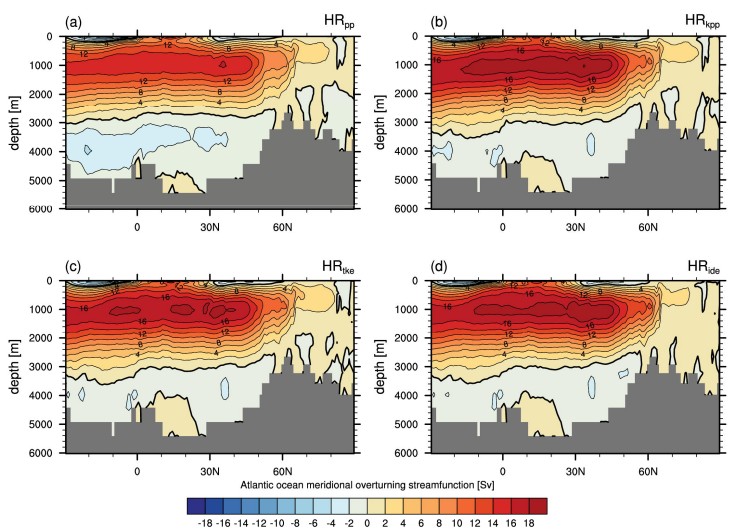

**Figure 13.** Eulerian stream function ($\mathrm{Sv} = 10^6\,\mathrm{m}^3\,\mathrm{s}^{-1}$) of the AMOC in depth space simulated by MPI-ESM1.2 (a) $\mathrm{HR}_{pp}$, (b) $\mathrm{HR}_{kpp}$, (c) $\mathrm{HR}_{tke}$, and (d) $\mathrm{HR}_{ide}$.

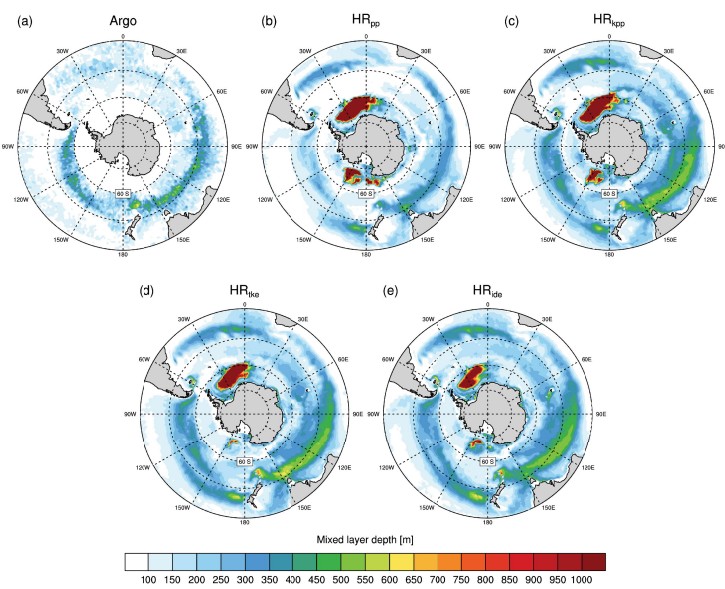

**Figure 14.** Time-averaged mixed layer depths (m) in September calculated (a) from $1° \times 1°$ Argo float data (Holte et al., 2017) (mean September from 2000 to 2018) by the density threshold method ($\sigma_t = 0.03 \, \mathrm{kg \, m^{-3}}$) and from MPI-ESM1.2 (b) $\mathrm{HR_{pp}}$, (c) $\mathrm{HR_{kpp}}$, (d) $\mathrm{HR_{tke}}$, and (e) $\mathrm{HR_{ide}}$. Note that there are no Argo data south of $60°$ S.

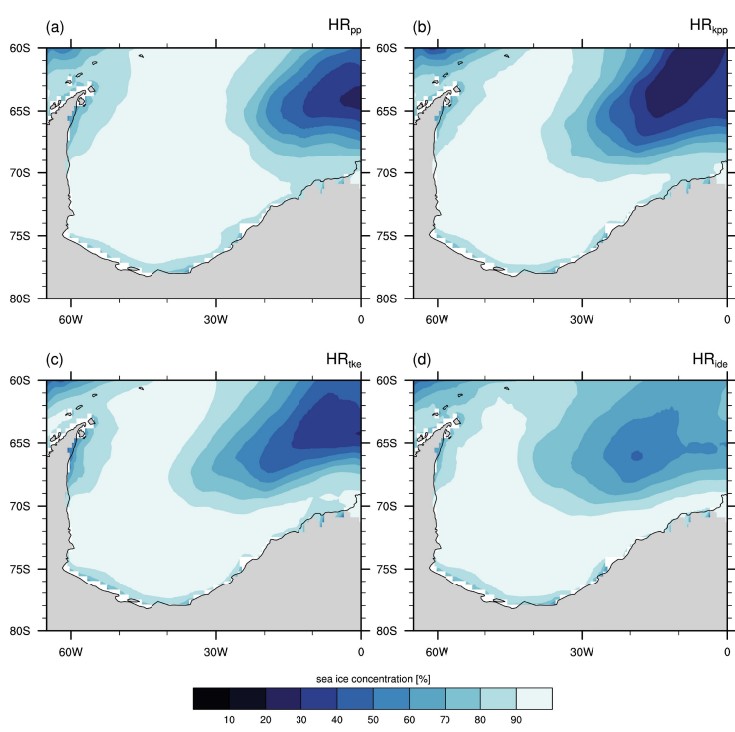

**Figure 15.** Time-averaged sea ice concentration in September in the Weddell Sea simulated by MPI-ESM1.2 (a) HR$_{pp}$, (b) HR$_{kpp}$, (c) HR$_{tke}$, and (d) HR$_{ide}$.





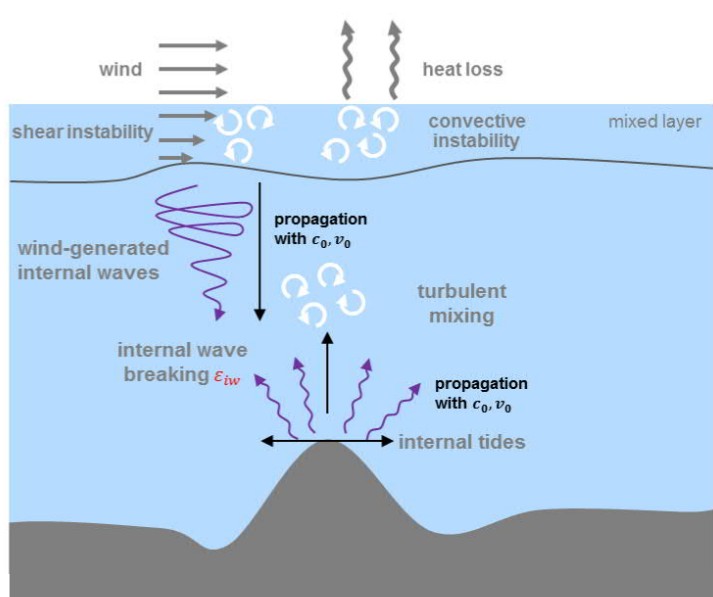

**Figure A1.** Schematic of the combined TKE+IDEMIX scheme used in HR$_{ide}$. Processes parameterised in TKE: away from strong currents, shear and buoyancy instability (convection) is largest near the surface (grey arrows), causing strong mixing in the mixed layer (white eddy symbols). Processes parameterized by IDEMIX: below the mixed layer, internal waves are either induced by fluctuating wind stress or by interactions of barotropic tides with orographic features (violet arrows). The internal waves are propagating into the interior ocean (black arrows), where they eventually break and dissipate (white eddy symbols).





**Table A1.** Resulting $p$-values from Welch's two-sided t-tests ($\alpha = 0.05$ and $n = 20$) for testing the differences in mean overflow volumes of Denmark Strait overflow (DSOW), in the Faroe-Shetland Channel (FSC), and in the Faroe Bank Channel (FBC).

| | Experiment | $HR_{pp}$ | $HR_{kpp}$ | $HR_{tke}$ | $HR_{ide}$ |
|---|---|---|---|---|---|
| **DSOW** | $HR_{pp}$ | - | $0.0^*$ | $0.0^*$ | $0.0^*$ |
| | $HR_{kpp}$ | - | - | 0.48 | 0.52 |
| | $HR_{tke}$ | - | - | - | 0.13 |
| | $HR_{ide}$ | - | - | - | - |
| **FSC** | $HR_{pp}$ | - | 0.23 | $0.0^*$ | 0.52 |
| | $HR_{kpp}$ | - | - | $0.02^*$ | 0.50 |
| | $HR_{tke}$ | - | - | - | $0.0^*$ |
| | $HR_{ide}$ | - | - | - | - |
| **FBC** | $HR_{pp}$ | - | 0.72 | $0.0^*$ | $0.0^*$ |
| | $HR_{kpp}$ | - | - | $0.0^*$ | $0.01^*$ |
| | $HR_{tke}$ | - | - | - | $0.0^*$ |
| | $HR_{ide}$ | - | - | - | - |

To achieve a power $(1\text{-}\beta)$ of $80\%$ with $n = 20$ and $\alpha = 0.05$, the minimum effect size $d = |\mu_1 - \mu_2| / \sqrt{(sd_1^2 + sd_2^2)/2}$ is about 1.0. For instance, a pooled standard deviation of e.g. $0.1\,\mathrm{Sv}$ would correspond to a minimum mean difference of $|\mu_1 - \mu_2| = 0.1\,\mathrm{Sv}$.