# Peer review of "Comparison of ocean vertical mixing schemes in the Max Planck Institute Earth System Model (MPI-ESM1.2)"

_Geoscientific Model Development, 2020_

## Referee Comment (RC1) · Anonymous Referee #1 · 30 Sep 2020

The paper compares the effects of four different ocean vertical mixing schemes provided by the Community Vertical Mixing (CVMIX) library on the mean state simulated by the coupled model MPI-ESM1.2. Used are the Pacanowski and Philander (PP), K-profile (KPP), turbulent kinetic energy (TKE) vertical mixing schemes as well as a prognostic schemes for internal wave energy and its dissipation (IDEMIX) which is combined with TKE. The author addresses temperature, salinity, and vertical mixing differences of the different vertical mixing schemes on a global scale but also regionally for crucial areas of the ocean, e.g. Fram Strait, Arctic Ocean, subpolar North Atlantic, Southern Ocean and Weddel Sea. As a summary, none of the presented vertical mixing schemes can be claimed superior on a global scale since they all produce

quite similar patterns. Only on a very regional scale some vertical mixing schemes can be favored of the other. Here, the more realistic and energetically consistent TKE+IDEMIX, with a more heterogeneous vertical mixing pattern, improves the circulation in the Nordic Seas and Fram-Strait reducing the bias of the Atlantic water layer in the Arctic Ocean. To my knowledge, the here presented work is novel to the MPI-ESM an general modeling community. The author presents quite well the biases shown by the different mixing schemes and attributes their causes especially on the regional scale. I would therefor recommend that the paper is accepted after some minor revision.

Comments: Abstract, line 3: The abbreviations for PP, KPP and TKE should be already made clear here.

1. Introduction: The author mentions the CVMIX library in the connection with TKE and IDEMIX it maybe should be made clearer that to this point neither TKE or IDEMIX are yet part of the CVMIX library, they just use its infrastructure routines and might join the project officially at some point.

1. Introduction: If I understood well, for PP vertical mixing, the MPI-ESM original PP implementation (which I guess is quite tuned) is used, not the CVMIX PP vertical mixing, right ? Reading the introduction from line 25 onward one might get a little bit miss leaded. It could be of benefit to clarify a bit more what at the end has been used from CVMIX. Furthermore, for my own interest, was the CVMIX PP parameterisation implemented into MPI-ESM and has there been also a comparison between the original PP and CMVIX PP implementation.

1. Introduction: Although PP and KPP are very common vertical mixing schemes, often described and widely used in the ocean modeling community, TKE is a bit more exotic but also not completely novel. It would be nice to have some more information about what has been done with TKE by others, for example in the NEMO community (e.g. Breivik, Ø. et. al 2015, Surface wave effects in the NEMO ocean

model: Forced and coupled experiments, J. Geophys. Res. Oceans, 120, 2973–2992, doi:10.1002/2014JC010565.)

1. Introduction, line 66: Despite the latter but because of ... Please reformulate this sentence.

1. Introduction, line 69: In section 2 we briefly... Please reformulate this sentence.

2. Model description:, line 89: ...Community Vertical Mixing (CVMIX) . . . replace with CVMIX. . . (Abbreviation already defined in introduction)

2. Model description:, line 92: . . . (TKE: Gaspar et al., 1990. . .replace with ...TKE (Gaspar et al., 1990. . .

2. Model description:, line 94: . . . because both schemes rely on . . . replace with ... because TKE and IDEMIX rely both on. . .

2.1 Experiments: Does MPI-ESM show any differences in the spin-up behavior (model drift, convergence,...) when using different vertical mixing scheme. Are there any differences in temporal evolution of quantities (e.g. AMOC, overflow, . . .).

3.1 Spatial distribution of the vertical diffusivity: line 124: ... where N is large and a large K in the high-latitude ocean where N is small . . . replace with . . . where N is positive and a large K in the high-latitude ocean where N is negative...

3.2 Sea surface temperature and salinity bias: line 138: . . . generate biases, the causes of which are often complex. . . . replace with . . . generate biases, whose causes are often complex. ...

3.2 Sea surface temperature and salinity bias: line 138: . . . the resolution, discretisation, and parameterisation of . . . replace with . . . the resolution, the vertical discretisation, and the parameterisation of . . .

3.2 Sea surface temperature and salinity bias: line 140: . . . with vertical mixing being just on complex process . . . replace with . . . with vertical mixing being just on of the

complex processes . . .

3.2 Sea surface temperature and salinity bias: line 147: The North Atlantic SST is sensitive. . . Please reformulate this sentence.

3.2 Sea surface temperature and salinity bias: line 153: . . . probably due to increased inflow from the Mc Kenzie River. Is this an educated guess or are their any proves for it in the model?

3.3.1 Horizontal maps of hydrographic biases: line 156: Why using the 740m depth layer?

3.3.1 Horizontal maps of hydrographic biases: line 173: Probably, using IDEMIX reduces the vertical mixing in the Mediterranean Sea and especially near the overflow sill ...  Is this statement no rather counter-intuitive?  Would one not expect the under IDEMIX, there should be more vertical mixing along the continental slopes of the Mediterranean and the outflow area?

4.1.1 Fram Strait: line 215: Wekerle, C., Wang, Q., von Appen, W.-J., Danilov, S., Schourup-Kristensen, V., & Jung, T. (2017). Eddy-resolving simulation of the Atlantic Water circulation in the Fram Strait with focus on the seasonal cycle. Journal of Geophysical Research: Oceans, 122, 8385–8405. https://doi.org/10.1002/2017JC012974 should be cited here as well.

4.1.1 Fram Strait: line 215: . . . recent studies indicate a third pathway of the WSC ... From the context before must it not be ...a fourth pathway...

4.1.2 Arctic Ocean: line 262: . . .Turbulence in the quiescent interior Arctic ocean . . . replace with . . . Turbulence in the interior Arctic ocean...

4.2.1 Convection and mixed layer depths: line 304: Maybe I oversaw it but is somewhere said which MLD definition is used?  Also regarding Fig. 10 and Fig. 14, the colorbar seems to be cut of at a 1000m. It would be nice if at least the text could mention the actual simulated maximum value of MLD also as general information for the

broader modeling community.

4.2.2 Overflows from the Nordic Seas: line 357: ... the FSC overflows are of about similar magnitude ... replace with ... the FSC overflows are of similar magnitude ...

Please also note the supplement to this comment:
https://gmd.copernicus.org/preprints/gmd-2020-202/gmd-2020-202-RC1-supplement.pdf

---

## Referee Comment (RC2) · Anonymous Referee #2 · 29 Oct 2020

In this manuscript the authors compared four different ocean vertical mixing schemes in MPI-ESM1.2 in 100-year long fully coupled simulations representing the conditions of 1950s. They focused their analysis on the mean state of important ocean variables, such as temperature, salinity, mixed layer depth and vertical eddy diffusivity and discussed the differences in these ocean vertical mixing schemes in explaining common biases in climate simulations, in both the global patterns and regional patterns at high latitudes. They found relatively small sensitivity of the simulated SST, SSS to the changes in the vertical mixing schemes but bigger sensitivity of the ocean interior. They conclude that 1) model resolution determines the global-scale bias pattern, and 2) vertical mixing schemes may play an important role for regional biases.

[Figure]

I think this is an interesting study. Comparing ocean vertical mixing schemes in a single modeling framework is also of practical use for the ocean and climate modeling communities. But I think there are some aspects this manuscript could be improved before being published.

General comments:

Throughout the manuscript, the authors are comparing the simulated results, which is the mean of the last 20 years of a 100-year long simulation under constant 1950s conditions, with the EN4 observation during the years 1945-1955, which is a state the simulations initialized from (or at least close to by nudging). I was wondering how sensible it is to use the word "biases" here to describe the differences, especially considering that the authors talks a lot about model biases reported in other studies in section 3 and 4 (which is good by the way). I just don't see how relevant the "biases" reported in this way are in terms of reflecting the "true" model biases in a historical climate simulation under transient greenhouse gas forcing. For example, do you expect the results of a "perfect" model to match the EN4 (1945-1955) observation in this experiment setup? I think it perhaps makes more sense to frame the discussion to focus more on the differences among the four simulations with different vertical mixing schemes and on which scheme, and in what ways, has the potential to fix the model biases reported in the literature, instead of targeting on a direct comparison with the observation, which I think will need more careful design of the simulations.

Related to the above comment, I think this manuscript could be improved by improving the clarity of the analyses in section 3 and 4. The thing I like about in these analyses is a summary of the relevant model biases reported in previous studies. However, I feel that the discussion of the simulation results itself is sometimes rather separated from these nice summary. I think the authors might want to be more specific in the reasoning and refer more frequently to the features in the figures in order to show what aspects of the different ocean vertical mixing schemes have the potential to fix the existing model biases reported in the literature. Sometimes I feel confused about which statement is

from the simulation results and which is from the literature.

Another thing I was hoping to see in this manuscript is some more insights of the differences among the four ocean vertical mixing schemes and more reasoning of how these differences in the schemes lead to differences in the simulation results. The authors discussed relatively more on the interior mixing below the surface mixed layer, which is quite simple especially for PP and KPP. But these scheme differ quite a lot in the mixed layer. For example, the implementation of KPP in this study used the same interior mixing as PP (according to Table 1), yet the results are often quite different between the KPP and PP simulations. It would be helpful if the authors could elaborate more on how the differences in the surface mixing contribute to the differences in the simulation results.

Specific comments:

L6: It is a bit unclear what you mean by "little sensitivity of the ocean surface", perhaps be more specific on what ocean surface variables and ocean interior variables, and be explicit on the sensitivity to changes in the ocean vertical mixing schemes.

L12: Are you comparing the effects of vertical mixing and the horizontal processes?

L13: How did you reach the first conclusion about the model resolution? Is the model resolution a focus of this study too?

L20: Temperature and salinity are active tracers

L20: "uptake" -> "ocean uptake"?

L23: Unclear statement. The complexity of a parameterization also depends on the physical and computational requirements in an ocean model. We could have a physically more favorable scheme based on our best understanding, but it could be too computationally expensive or not necessary for a simple model.

L26: I'm not sure if PP, perhaps even KPP, is "state-of-the-art". They are widely used

though.

L32: There are actually small modifications to the implementations of a certain scheme, such as KPP, happening throughout the time due to practical reasons, e.g., Appendix A of Danabasoglu et al., 2006

L33: Numerical implementation based on the same principles may also matter. See, e.g., the comparison of the CVMix version of KPP and ROMS version in Li et al., 2019.

L34-35: "schemes provide either direct vertical profiles" -> Perhaps something like "schemes diagnose vertical profiles of ... from surface forcing and background fields"

L36-37: I believe these schemes also only provides eddy diffusivity and viscosity when implemented in an ocean circulation model, not the fluxes. The key difference is that both PP and KPP are diagnostic which assume equilibrium with the current forcing and background state, whereas second-order schemes have memory of previous states.

L42: Briefly introducing ECHAM6.3 for those reader who are not familiar with this model? For example, "ECHAM6.3, the atmosphere model developed at ...,"

L42: What do you mean by unstable? Does the AMOC shut down?

L48: "it depends" -> "depending"?

L55: I think Olbers and Eden, 2013 is a more appropriate reference here.

L57: "not only represents" -> "represents not only"?

L63-64: Be more specific on "a minor effect on the climate state"?

L66: Confusing, please rephrase.

L96: Delete "control"?

L113: What do you mean by "unbiased effects"?

L114-115: If the coupling and feedbacks from the atmosphere are not discussed, why

not using the OMIP protocol to force the ocean model with atmospheric data? What is the advantages of using coupled runs here when comparing the four ocean vertical mixing schemes?

L119: "(section 4)" -> "in section 4"?

L121: Without being more specific focusing on your results, this statement is certainly not true, even for the ocean interior and excluding the effects of deep convection, which I assume you meant here. The vertical diffusivity is affected by many processes (such as internal waves, which depends on both the bathymetry as well as the surface forcing) and background state such as the stratification. A constant background diffusivity in PP and KPP is a simplification. You might want to rephrase, for example, to restrict it to the simple parameterizations of PP and KPP. You might also want to be specific that you are talking about the vertical diffusivity in the ocean interior away from the surface and bottom boundaries.

L151-153 and in Fig 3: The difference of SSS in the Arctic appears substantial (especially between panels a, b and c, d). You might want to elaborate more on the possible causes. For example, how the differences among the four schemes lead to the significantly different SSS. Does the simulated sea ice change a lot?

L153: I'm confused – Isn't the vertical mixing scheme the only difference among the four simulations? What do you mean by increased river inflow?

L168-170: It might just be due to the different vertical distribution of the salinity resulting from different vertical mixing. Does the horizontal flow change a lot among the four simulations, in order to support your hypothesis of stronger inflow of saline water from the Indian Ocean?

L181: What do you mean by "above named currents and water masses"?

L183-188: Again, you might want to support your hypotheses by more clear reasoning of how changing the vertical mixing results in the differences in, e.g., the inflow from the

[Figure]

Indian Ocean, and the overflow water at 60N and the MOW, thereby the temperature distribution.

L190: Perhaps "the next sections" -> "this section"?

L199: "background mixing value" -> "background eddy diffusivity and viscosity"?

L210-216: Comments on how these features are simulated in the four simulations? What are the conclusions from Fig. 7?

L222, 224: Fig 11 and Fig 12 are used before Fig 9 and Fig 10 without introduction?

L229-243: Again, how are these features simulated in the four simulations here? This review seems to be separated from the analysis of the simulation results.

L262: What do you mean by "close to the limit of measurement"?

L264-265: You mean the cell averaged wind stress? Is the ice stress accounted for here?

L285-286: Does this result in, say, more realistic sea ice results?

A question to most of the maps at a certain depth: how was the depth chosen for each map?

L317: Why not using the same density threshold?

L324-325: Do you mean comparing with HR_{tke} here? How is the internal waves affecting the surface mixed layer?? Is it due to more frequent convection as a result of a stronger interior mixing (due to internal waves) and thus weaker stratification?

L337: The disappearance of the bottom cell is pronounced...

L340-341: Would you be more specific about this statement? I can see quite some differences between HR_{tke} and HR_{kpp} in many of your figures, e.g., MLD and AMOC

L408-409: Rephrase? Do you mean the difference between the three are small?

L410: Delete "but"?

L486 and Eq(A6): Why a change of notation here (removing the prime)?

L498: 0.7 is significantly bigger than the suggestion of 0.3 in Large et al., 1994 and many implementations of KPP via CVMix in other models. Any comments on why this value is used?

Appendix A2: Since the KPP scheme is well documented in the literature, especially in the CVMix documentation, I would suggest the authors to consider the necessity of showing all the details here. I think it would be much cleaner for the reader if you only briefly introduce KPP and highlight the different configurations than the default in CVMix when implemented in MPI-ESM1.2.

L534-537: What are the values for these parameters that are used in this study?

L564: Combine with the previous sentence and delete "When TKE is used alone without being coupled to IDEMIX"?

References:

Danabasoglu, G., Large, W. G., Tribbia, J. J., Gent, P. R., Briegleb, B. P., & McWilliams, J. C. (2006). Diurnal coupling in the tropical oceans of CCSM3. Journal of Climate, 19(11), 2347–2365. https://doi.org/10.1175/JCLI3739.1
* * *

---

## Author Comment (AC1) · 29 Jan 2021

Please see the attached pdf.

Please also note the supplement to this comment:
https://gmd.copernicus.org/preprints/gmd-2020-202/gmd-2020-202-AC1-supplement.pdf
* * *

---

## Author Response (AR1)

We thank the reviewers for their insightful and helpful comments, which we think have greatly improved our manuscript. In light of all these helpful comments, we have restructured the paper, adding new sections e.g. on sea ice, removing or streamlining sections, and in particular we have added analyses for the atmosphere. The main conclusion of the paper now focuses on the process chain that evolves when changing the vertical mixing to KPP or TKE.

In the following, we respond sequentially to all their comments.

**Reviewer 1:**

**Abstract, line 3: The abbreviations for PP, KPP and TKE should be already made clear here.** We introduced the PP, KPP and TKE scheme now in the Abstract.

**1. Introduction: The author mentions the CVMIX library in the connection with TKE and IDEMIX it maybe should be made clearer that to this point neither TKE or IDEMIX are yet part of the CVMIX library, they just use its infrastructure routines and might join the project officially at some point.**
A: This is correct, only the infrastructure of CVMix was used. We have corrected this in the Introduction, stating also that both TKE and IDEMIX are not yet official part of CVMix.

**1. Introduction: If I understood well, for PP vertical mixing, the MPI-ESM original PP implementation (which I guess is quite tuned) is used, not the CVMIX PP vertical mixing, right? Reading the introduction from line 25 onward one might get a little bit miss leaded. It could be of benefit to clarify a bit more what at the end has been used from CVMIX. Furthermore, for my own interest, was the CVMIX PP parameterisation implemented into MPI-ESM and has there been also a comparison between the original PP and CMVIX PP implementation.**
A: Correct, the PP mixing we compare here is the internal version of MPI-ESM, which differs from the original formulation of Pacanowski & Philander (1981) by adding an additional wind-induced mixing term. MPI-ESM was tuned using this modified version. Concerning the last point, the CVMix-PP is technically implemented in MPI-ESM but was never used. We make clear now that we do not use the version that comes with CVMix.
From our point of view, it is not useful to use the PP version of CVMix, as MPI-ESM would never be used with the CVMix PP scheme, as it was found that the original formulation lacked mixing due to wind stress near the surface. Therefore, we have not done a comparison with the CVMix PP scheme, but for other modelling groups this would of course be an option.

**1. Introduction: Although PP and KPP are very common vertical mixing schemes, often described and widely used in the ocean modeling community, TKE is a bit more exotic but also not completely novel. It would be nice to have some more information about what has been done with TKE by others, for example in the NEMO community (e.g. Breivik, Ø. et. al 2015, Surface wave effects in the NEMO ocean model: Forced and coupled experiments, J. Geophys. Res. Oceans, 120, 2973–2992, doi:10.1002/2014JC010565.)**
A: We agree that TKE is probably less often used by the ocean modelling community. We have added a sentence in the Introduction: "Although KPP is probably the most widely used scheme in ocean models, TKE is also a frequent choice and is part of state-of-the-art ocean models, and for which also extensions such as Langmuir turbulence (Axell, 2002) or surface waves (Breivik et al., 2015) were developed."

**1. Introduction, line 66: Despite the latter but because of ... Please reformulate this sentence.**
A: We have simplified the sentence to "Due to these promising results, we compare …".

**1. Introduction, line 69: In section 2 we briefly... Please reformulate this sentence.**

A: We rephrased and split the sentence to: "We first give a brief overview of the model configuration in section 2, with more details about the vertical mixing schemes and the experiments we conducted. In section 3, we present the results of the comparison for the global ocean and in section 4 for the regional ocean. Section 5 presents effects of the mixing scheme in the atmosphere. Finally, we conclude in section 6.".

**2. Model description:, line 89: ...Community Vertical Mixing (CVMIX) … replace with CVMIX… (Abbreviation already defined in introduction)**
A: Corrected.

**2. Model description:, line 92: … (TKE: Gaspar et al., 1990 … replace with ...TKE (Gaspar et al., 1990:**
A: Corrected. We have also added here that the TKE and IDEMIX schemes are not yet officially available from CVMix.

**2. Model description:, line 94: … because both schemes rely on … replace with ... because TKE and IDEMIX rely both on…**
A: Corrected.

**2.1 Experiments: Does MPI-ESM show any differences in the spin-up behavior (model drift, convergence,...) when using different vertical mixing scheme. Are there any differences in temporal evolution of quantities (e.g. AMOC, overflow, …).**
A: We checked the time series of AMOC (see Fig.1 below). While the AMOC is rather constant or weakly declining, the AMOC strengthens within the first 10-20 years or so with KPP and TKE, residing at a higher value thereafter. With IDEMIX (HRide) the AMOC is even lower as with PP in the first half of the simulation but quickly rises in the beginning years of the second half (after year 2000) to values that are comparable with KPP and TKE. Therefore, we conclude that there is a different temporal behaviour, but in the last 20 model years that we analyse, the AMOC is rather stable.

[Figure]

*Fig.1: 5-year running mean time series of AMOC at 26°N from 1950 to 2050 in MPI-ESM1.2-HR.*

**3.1 Spatial distribution of the vertical diffusivity: line 124: ... where N is large and a large K in the high-latitude ocean where N is small … replace with … where N is positive and a large K in the high-latitude ocean where N is negative...**
A: Corrected.

**3.2 Sea surface temperature and salinity bias: line 138: … generate biases, the causes of which are often complex. … replace with … generate biases, whose causes are often complex. ...**
A: Corrected.

**3.2 Sea surface temperature and salinity bias: line 138: … the resolution, discretisation, and parameterisation of … replace with … the resolution, the vertical discretisation, and the parameterisation of …**
A: Corrected.

**3.2 Sea surface temperature and salinity bias: line 140: … with vertical mixing being just on complex process … replace with … with vertical mixing being just on of the complex processes …**
A: Corrected.

**3.2 Sea surface temperature and salinity bias: line 147: The North Atlantic SST is sensitive … Please reformulate this sentence.**
A: We have rephrased the sentence to "By using a vertical mixing scheme other than PP, we find that the SST cold bias in the North Atlantic is reduced (Fig.2b-d)."

**3.2 Sea surface temperature and salinity bias: line 153: … probably due to increased inflow from the Mc Kenzie River. Is this an educated guess or are their any proves for it in**
A: Yes it is an educated guess. By looking at the sea surface salinity bias in the Arctic (see Fig. 2 below), we noted that simulations with PP and KPP produce a positive salinity bias that stretches from the Northwestern Territories of Canada and Alaska over the Beaufort Sea to the north of Ellesmere Island. However, it is not clear whether this is related to river runoff or to the formation of sea ice in general, which is lower with TKE and IDEMIX and would result in less brine rejection. But we do not have a satisfying answer to this yet.

[Figure]

*Figure 2: Sea surface salinity bias (MPI-ESM1.2 minus EN4) in the Arctic Ocean.*

**3.3.1 Horizontal maps of hydrographic biases: line 156: Why using the 740m depth layer?**

A: The depth of 740m is the depth of a model layer that was also used in Gutjahr et al. (2019). We selected this layer here for a better comparison with that study, but the model biases are very similar to e.g. a depth of 700m.

**3.3.1 Horizontal maps of hydrographic biases: line 173: Probably, using IDEMIX reduces the vertical mixing in the Mediterranean Sea and especially near the overflow sill ... Is this statement no rather counter-intuitive? Would one not expect the under IDEMIX, there should be more vertical mixing along the continental slopes of the Mediterranean and the outflow area?**

A: Indeed, there is higher mixing at the overflow sill and downstream in the Gulf of Cádiz. However, over the abyssal plains further to the west, die vertical diffusivity is one magnitude less in HRide (see Fig.3d below). We make clear that we speculate that this reduced mixing reduces the mixing with ambient water but also state that there could be other factors, such as the near-surface wind field and net evaporation over the Mediterranean basin. We revised the manuscript to make this clear.

[Figure]

*Fig 3: Vertical diffusivity log10(K) (m²s⁻¹) at a depth of 1020m in the Gulf of Cádiz.*

**4.1.1 Fram Strait: line 215: Wekerle, C., Wang, Q., von Appen, W.-J., Danilov, S., Schourup-Kristensen, V., & Jung, T. (2017). Eddy-resolving simulation of the Atlantic Water circulation in the Fram Strait with focus on the seasonal cycle. Journal of Geophysical Research: Oceans, 122, 8385–8405. https://doi.org/10.1002/2017JC012974 should be cited here as well.**

A: Yes of course, we have added the reference.

**4.1.1 Fram Strait: line 215: … recent studies indicate a third pathway of the WSC ... From the context before must it not be ...a fourth pathway...**

A: You are right, this paragraph is misleading. We have revised the whole section on Fram Strait and the Atlantic water layer and removed information that is not necessary to understand our results. In particular, since we do not analyse the branches of the AW itself, we removed much of the details about these currents.

**4.1.2 Arctic Ocean: line 262: … Turbulence in the quiescent interior Arctic ocean … replace with … Turbulence in the interior Arctic ocean...**

A: We have changed the sentence to "Although largely unknown, sparse observations indicate that turbulence in the Arctic Ocean is typically weak.".

**4.2.1 Convection and mixed layer depths: line 304: Maybe I oversaw it but is somewhere said which MLD definition is used? Also regarding Fig. 10 and Fig. 14, the colorbar seems to be cut of at a 1000m. It would be nice if at least the text could mention the actual simulated maximum value of MLD also as general information for the broader modeling community.**

A: We used the density threshold of 0.01 kg/m³ in the subpolar North Atlantic and 0.03 kg/m³ in the Southern Ocean. However, we replaced Fig.11 (now Fig. 12) using now the same density threshold (0.03 kg/m³) for ARGO and model data in both hemispheres. We also adjusted the colourbar to better distinguish very deep mixed layers (also for MLD in the Southern Ocean, now Fig. 16).

**4.2.2 Overflows from the Nordic Seas: line 357: … the FSC overflows are of about similar magnitude … replace with … the FSC overflows are of similar magnitude ...**

A: Corrected.

**Reviewer 2:**

**General comments:**

**I was wondering how sensible it is to use the word "biases" here to describe the differences, especially considering that the authors talks a lot about model biases reported in other studies in section 3 and 4 (which is good by the way). I just don't see how relevant the "biases" reported in this way are in terms of reflecting the "true" model biases in a historical climate simulation under transient greenhouse gas forcing. For example, do you expect the results of a "perfect" model to match the EN4 (1945-1955) observation in this experiment setup? I think it perhaps makes more sense to frame the discussion to focus more on the differences among the four simulations with different vertical mixing schemes and on which scheme, and in what ways, has the potential to fix the model biases reported in the literature, instead of targeting on a direct comparison with the observation, which I think will need more careful design of the simulations.**
A: There are two reasons why we decided to compare the biases of ocean temperature and salinity, which are practically deviations from the initial state, since all simulations were initialized with EN4 around 1950. First, the simulations originate from the EU Horizon-2020 PRIMAVERA project. This project pursued two strands of research questions: the effect of horizontal resolution and physical improvements on climate simulations. While the first question - effect of horizontal resolution - was dealt with in the work of Putrasahan et al. (2019) and Gutjahr et al. (2019), the present paper pursues the second question – the effect of physical improvements in our MPI-ESM1.2 simulations. It is also for the HighResMIP protocol that we used coupled simulations.
And second, although we performed also historical and scenario simulations with MPI-ESM1.2-HR model using PP/KPP, we did not with TKE and IDEMIX. However, we noticed that the systematic model biases are similar in our control and historical simulations (PP/KPP) and are also similar to previous studies with MPI-ESM, independent on the GHG forcing. Therefore, we think our study design is comparable to biases in historical simulations and that these are mainly related to insufficient resolution or physical parameterisations. Besides, substracting the observed mean state from the simulated mean is a linear operation that does not change the results or conclusions compared to inter-model comparisons.
We agree that there might be a better design to compare to observations, but we also note that a comparison with gridded observational or reanalysis data is never perfect. We hope that this explanation justifies our study design, which is mainly determined by the available simulations, the HighResMIP protocol and the initialisation data.

**Related to the above comment, I think this manuscript could be improved by improving the clarity of the analyses in section 3 and 4. The thing I like about in these analyses is a summary of the relevant model biases reported in previous studies. However, I feel that the discussion of the simulation results itself is sometimes rather separated from these nice summary. I think the authors might want to be more specific in the reasoning and refer more frequently to the features in the figures in order to show what aspects of the different ocean vertical mixing schemes have the potential to fix the existing model biases reported in the literature. Sometimes I feel confused about which statement is from the simulation results and which is from the literature.**
A: We have revised all sections of the manuscript, thereby removing information that is not relevant for our study. We hope that the manuscript is now easier to read and that confusing passages are more comprehensible. As this paper serves as an overview, we cannot explain all differences we see in the model. To identify a specific term of the vertical mixing parameterisation would require additional analysis, which, however, go beyond the intention of our manuscript.

**Another thing I was hoping to see in this manuscript is some more insights of the differences among the four ocean vertical mixing schemes and more reasoning of how these differences in the schemes lead to differences in the simulation results. The authors discussed relatively more on the interior mixing below the surface mixed layer, which is quite simple especially for PP and KPP. But these**

**scheme differ quite a lot in the mixed layer. For example, the implementation of KPP in this study used the same interior mixing as PP (according to Table 1), yet the results are often quite different between the KPP and PP simulations. It would be helpful if the authors could elaborate more on how the differences in the surface mixing contribute to the differences in the simulation results.**

A: We agree that individual aspects of the mixing schemes could be discussed in more detail. However, often a change in a model bias is composed of complex interactions, which is probably not possible to disentangle with our study design. We aim here at a first order comparison to what can be expected in terms of model biases when the vertical mixing scheme is exchanged in a coupled climate model. We tried to give reasons for different model responses where possible, but for more detailed explanations idealized simulations might be necessary.

**Specific comments:**

**L6: It is a bit unclear what you mean by "little sensitivity of the ocean surface", perhaps be more specific on what ocean surface variables and ocean interior variables, and be explicit on the sensitivity to changes in the ocean vertical mixing schemes.**

A: We have revised to Abstract and are more specific about the effects we find from exchanging the ocean vertical mixing scheme. We have also included results for the atmosphere now and revised our statement, describing now that the SSTs warm in

**L12: Are you comparing the effects of vertical mixing and the horizontal processes?**

A: We referred here to both: using TKE+IDEMIX reduces the warm bias of the Atlantic water layer in the Arctic Ocean to a similar extent as in an eddy-resolving (0.1°) simulation we did in an earlier study (Gutjahr et al., 2019) that followed the same protocol. However, we have rephrased the paragraph (and section).

**L13: How did you reach the first conclusion about the model resolution? Is the model resolution a focus of this study too?**

A: The biases in salinity and temperature persist in all simulations performed with MPI-ESM1.2-HR and earlier simulations with MPI-ESM-LR. However, we could show that by using a higher resolution ocean model (0.1°) that many of these biases are diminished (Gutjahr et al., 2019). Since changing the vertical mixing scheme does not reduce these biases (e.g. associated with the Agulhas or the Mediterranean Overflow), we conclude that these mainly result from a too coarse horizontal resolution in these areas. We agree that we do not directly compare with model resolution and have removed this conclusion.

**L20: Temperature and salinity are active tracers**

A: We have corrected this.

**L20: "uptake" -> "ocean uptake"?**

A: Corrected.

**L23: Unclear statement. The complexity of a parameterization also depends on the physical and computational requirements in an ocean model. We could have a physically more favorable scheme based on our best understanding, but it could be too computationally expensive or not necessary for a simple model.**

A: We have rephrased this sentence to: "The complexity of these parameterisations varies in dependence of our understanding, application, and available resources".

**L26: I'm not sure if PP, perhaps even KPP, is "state-of-the-art". They are widely used though.**

A: We have removed "state-of-the-art" from the sentence.

**L32: There are actually small modifications to the implementations of a certain scheme, such as KPP, happening throughout the time due to practical reasons, e.g., Appendix A of Danabasoglu et al., 2006**
A: We have added this remark to the paragraph.

**L33: Numerical implementation based on the same principles may also matter. See, e.g., the comparison of the CVMix version of KPP and ROMS version in Li et al., 2019.**
A: We have added this remark to the paragraph.

**L34-35: "schemes provide either direct vertical profiles" -> Perhaps something like "schemes diagnose vertical profiles of ... from surface forcing and background fields"**
A: We have adjusted the sentences to "In the ocean surface boundary layer, schemes diagnose vertical profiles of scalar mixing diffusivity and viscosity from surface forcing and background fields, such as in the PP scheme (Pacanowski and Philander, 1981) or in the K-profile parameterisation KPP; Large et al., 1994). Second order schemes (Mellor and Yamada, 1982}, such as the TKE scheme (Gaspar, 1990), contain in addition to the mean quantities also prediction equations for higher order moments, i.e. for variance and covariance terms of heat and momentum."

**L36-37: I believe these schemes also only provides eddy diffusivity and viscosity when implemented in an ocean circulation model, not the fluxes. The key difference is that both PP and KPP are diagnostic which assume equilibrium with the current forcing and background state, whereas second-order schemes have memory of previous states.**
A: That is correct. We have corrected the sentences, see comment above.

**L42: Briefly introducing ECHAM6.3 for those reader who are not familiar with this model? For example, "ECHAM6.3, the atmosphere model developed at ...,"**
A: We have added the information as you suggest.

**L42: What do you mean by unstable? Does the AMOC shut down?**
A: Yes, we referred to a slowing down of the AMOC when ECHAM6.3 is used with a T255 resolution in combination with the PP scheme and a 0.4° ocean. We have clarified the sentences to "However, recent experiments with a higher-resolution (T255 or ~50 km) version of ECHAM6.3, the atmospheric model developed at MPI-M, resulted in a collapse of the AMOC and icing of the Labrador Sea (Putrasahan et al., 2019). By replacing PP with KPP, however, Gutjahr et al. (2019) showed that a stable AMOC is maintained."

**L48: "it depends" -> "depending"?**
A: Corrected.

**L55: I think Olbers and Eden, 2013 is a more appropriate reference here.**
A: That is correct, we removed Eden et al. (2014) here.

**L57: "not only represents" -> "represents not only"?**
A: Corrected.

**L63-64: Be more specific on "a minor effect on the climate state"?**
A: We rephrased the paragraph and are more specific about the results from Nielsen et al. (2018): "Using IDEMIX in coupled simulations, Nielsen et al. (2018) report only a minor effect on the sea surface temperature. However, they demonstrate reduced thermocline diffusivities with IDEMIX, which leads to a sharper and shallower thermocline, because less heat is mixed downwards. Although IDEMIX produces colder temperature within the first 1000 m of their simulations, at mid-depth the temperatures are in better agreement with observations."

**L66: Confusing, please rephrase.**
A: We rephrased to "Due to these promising results, we compare the effect of IDEMIX with the other mixing schemes of MPI-ESM1.2 and analyse regions that are most sensitive to IDEMIX on the typical time scale of 100 years for climate simulations."

**L96: Delete "control"?**
A: We removed "control" from the sentence.

**L113: What do you mean by "unbiased effects"?**
A: We rephrased to "As recommended in this protocol, the model was not retuned to obtain isolated effects from changing the ocean vertical mixing scheme."

**L114-115: If the coupling and feedbacks from the atmosphere are not discussed, why not using the OMIP protocol to force the ocean model with atmospheric data? What is the advantages of using coupled runs here when comparing the four ocean vertical mixing schemes?**
A: We agree and have added analysis for the atmosphere. Indeed we found warmer extratropics in the northern hemisphere with TKE(+IDEMIX) and warmer temperatures in almost all of troposphere with KPP. We introduced a new section (now section 5) that shows results from basic quantities in the atmosphere. Given these results, we revised the Conclusions and Abstract sections and describe a consistent picture that emerged when using a mixing scheme other than PP. In brief, KPP and TKE enhance the deep convection and hence the overflows in the subpolar NA and Nordic Seas. The roughly 10% higher overflow volumes contribute to a stronger and deeper upper cell of the AMOC. Further, the inflow from the Indian to the South Atlantic is increased. A stronger upper cell of the AMOC transports more heat and salt northwards leading to warmer temperatures in the SPNA and Nordic Seas (which is why the sea ice edge retreats) and the higher salinity maintains the enhanced convection. Warmer SSTs imprint on the atmosphere, which in turn warms. Depending on whether only the extratropics warm (TKE) or the whole troposphere (KPP), the meridional gradients weakens and, via the thermal wind relation, also the northern hemisphere jet stream (TKE).

**L119: "(section 4)" -> "in section 4"?**
A: Corrected.

**L121: Without being more specific focusing on your results, this statement is certainly not true, even for the ocean interior and excluding the effects of deep convection, which I assume you meant here. The vertical diffusivity is affected by many processes (such as internal waves, which depends on both the bathymetry as well as the surface forcing) and background state such as the stratification. A constant background diffusivity in PP and KPP is a simplification. You might want to rephrase, for example, to restrict it to the simple parameterizations of PP and KPP. You might also want to be specific that you are talking about the vertical diffusivity in the ocean interior away from the surface and bottom boundaries.**
A: That is of course true and we implicitly were referring to the interior ocean, away from boundary currents, deep convection areas and mixed layer processes. We rephrased the sentence according to your suggestion to "Away from boundary currents, deep convection areas and the surface mixed layer, the vertical diffusivity K is approximately homogeneously distributed to leading order in the simulations with PP and KPP, which both use the simple constant background diffusivity of $K=1.05*10^{-5}$ m²s⁻² for parameterising internal wave breaking, as demonstrated exemplary for a model layer at intermediate depth of 1020 m (Fig. 1)."

**L151-153 and in Fig 3: The difference of SSS in the Arctic appears substantial (especially between panels a, b and c, d). You might want to elaborate more on the possible causes. For example, how the differences among the four schemes lead to the significantly different SSS. Does the simulated sea ice change a lot?**

A: We agree that the differences are substantial. We did replot the SSS bias of the Arctic Ocean in a stereographic projection (see Fig. 3 above) and plotted the sea ice thickness and the 15% contour of the sea ice concentration in comparison to PIOMAS (1979-2005) ice thickness and OSI SAF (1979-2005) ice concentration (and Fig.4 below, now Fig.4 in the manuscript). We note that 1) the sea ice in the Canada Basin and north of Greenland is too low in all simulations, but becomes lower with KPP, TKE and TKE+IDEMIX; 2) the ice edge is most extensive in HRpp, in particular in the Nordic Seas. The ice edge in the Nordic Seas retreats in HRkpp, HRtke and most so in HRide, which is related to warmer temperatures in the Nordic Seas with KPP and TKE. In summer (Fig. 5 below) the sea ice is also thinner with KPP and TKE. This indicates that the sea ice parameters, such as lead closure, might need retuning. Note that we have also added map of the ice thickness in the SH for September (now Fig. 5) in the manuscript. The ice is also thinner with KPP and TKE, indicating that the tuning of the sea ice module for PP is not optimal for KPP and TKE.

[Figure]

*Fig. 4: Average March sea ice thickness from (a) PIOMAS (1979-2005; Zhang and Rothrock, 2003) and (b-e) MPI-ESM1.2-HR. Overlain is the 15% sea ice concentration contour from PIOMAS (dark blue) and from the individual simulations (magenta).*

[Figure]

Fig. 5: As Fig. 4 but for September.

**L153: I'm confused – Isn't the vertical mixing scheme the only difference among the four simulations? What do you mean by increased river inflow?**
A: We were just speculating about the origin of the fresher SSS in the Arctic Ocean. It could be related to enhanced freshwater input by rivers. River runoff is not constant but calculated by a river routing scheme. However, it is also possible that the reduced ice production with TKE reduces the brine rejection. We do not have a definite answer for this difference in SSS.

**L168-170: It might just be due to the different vertical distribution of the salinity resulting from different vertical mixing. Does the horizontal flow change a lot among the four simulations, in order to support your hypothesis of stronger inflow of saline water from the Indian Ocean?**
A: You are right, we did not show transport volumes for the Agulhas of the simulations. From Fig. 6 below (Fig. 7 in the revised manuscript), there is a clear increased inflow from the Indian Ocean to the South Atlantic visible with KPP and TKE. This inflow is about 10 Sv stronger with TKE, about 15Sv with KPP. This correlates well with the strength of the AMOC, which is strongest with KPP. We have added these numbers to the manuscript: "Although all simulations show a similar salinity bias in the Agulhas region, we note a larger bias for HRkpp, HRtke, and HRide. This larger bias indicates a stronger inflow of warm and salty water from the Indian Ocean. In fact, the inflow is about 10 Sv stronger with TKE and about 15 Sv with KPP than the 40 to 50 Sv in HRpp (Fig. 7)."

[Figure]

*Fig. 6: Volume stream function (Sv) for the Agulhas region in (a) HRpp and the differences (experiment – HRpp) for (b) HRkpp, (c) HRtke, and (d) HRide.*

**L181: What do you mean by "above named currents and water masses"?**
A: We are sorry, this is misleading. We revised the entire section 4 and changed the introduction sentence to: "In this section, we discuss some regional areas in more detail, in particular the Atlantic Ocean, the Nordic Seas and Fram Strait, the Arctic Ocean, and the Southern Ocean."

**L183-188: Again, you might want to support your hypotheses by more clear reasoning of how changing the vertical mixing results in the differences in, e.g., the inflow from the Indian Ocean, and the overflow water at 60N and the MOW, thereby the temperature distribution.**
A: We have revised the entire manuscript to link all our findings in a consistent picture. See our answer to your question to L114-115 above. This connection of processes is now the main conclusion from the paper.

**L190: Perhaps "the next sections" -> "this section"?**
A: Corrected.

**L199: "background mixing value" -> "background eddy diffusivity and viscosity"?**
A: Corrected to "background diffusivity"

**L210-216: Comments on how these features are simulated in the four simulations? What are the conclusions from Fig. 7?**
A: We have removed most of the details about the currents from the section, thereby reducing it to

information that is necessary to understand the area. In the process we have also removed Fig. 7 and refer now only to the volume transport in the Nordic Seas (now Fig.12 in the manuscript).

**L222, 224: Fig 11 and Fig 12 are used before Fig 9 and Fig 10 without introduction?**
A: Due to new figures and rearrangements, the order of the figures has changed completely, but we made sure that the references in the text are also in the correct order.

**L229-243: Again, how are these features simulated in the four simulations here? This review seems to be separated from the analysis of the simulation results.**
A: See our answer for your comment on L210-216. We streamlined the section and hope that it reads now more clearly.

**L262: What do you mean by "close to the limit of measurement"?**
A: This was misleading and we have rephrased it to "Although largely unknown, sparse observations indicate that turbulence in the Arctic Ocean is typically weak (Rainville and Winsor, 2008; Fer, 2009).".

**L264-265: You mean the cell averaged wind stress? Is the ice stress accounted for here?**
A: Reduced is the Kw in the PP parameterization by (1-A), where A is the sea ice concentration (equation A2). For KPP, the vertical turbulent velocity scale reduces in the presence of sea ice (equation A9). We therefore revised the sentence to "The wind stress cannot act on the sea surface because of the insulating sea ice cover, which is why the effect of the wind stress on vertical mixing decreases quadratically with the sea ice concentration in the simulations with PP and KPP".

**L285-286: Does this result in, say, more realistic sea ice results? A question to most of the maps at a certain depth: how was the depth chosen for each map?**
A: It was chosen as one model level, which was also used in Gutjahr et al. (2019). We added maps for sea ice thickness (now Fig.4 and 5 in the manuscript) and compare the thickness with the PIOMAS reanalysis and the ice extent with satellite observations of the OSI SAF product. We find that the ice extent seems to be more realistic; in particular in the SPNA and Nordic Seas. The thickness, however, is too low in all simulations; most pronounced in the simulations with KPP and TKE. These results suggest that the tuning of the sea ice module with the PP scheme should be retuned for ice thickness when KPP or TKE is used.

**L317: Why not using the same density threshold?**
A: We have replaced Fig.11 (now Fig. 12) and are using now the same density threshold of 0.03 kg/m³ for the ARGO and model data. We have further adjusted the scale of the colourbar to take into account the deeper mixed layers and to allow better differentiation in the Labrador Sea. We also adjusted the colourbar in Fig. 12 and Fig. 16.

**L498: 0.7 is significantly bigger than the suggestion of 0.3 in Large et al., 1994 and many implementations of KPP via CVMix in other models. Any comments on why this value is used?**
A: Thank you for spotting this error. It was just a typing error; we used the default value of 0.3 for the critical Richardson number.

**Appendix A2: Since the KPP scheme is well documented in the literature, especially in the CVMix documentation, I would suggest the authors to consider the necessity of showing all the details here. I think it would be much cleaner for the reader if you only briefly introduce KPP and highlight the different configurations than the default in CVMix when implemented in MPI-ESM1.2.**
A: We agree in principal but we decided to keep the section on KPP in the manuscript for reasons of documentation. A user of MPI-ESM1.2 could now find and easier compare all information on the available options for vertical mixing parameterisation in this manuscript.

**L534-537: What are the values for these parameters that are used in this study?**
A: We have added the input fields for the surface and bottom boundary as Fig.A2 in the appendix.

**L564: Combine with the previous sentence and delete "When TKE is used alone without being coupled to IDEMIX"?**
A: We have corrected the sentence.

---

## Author Response (AR2)

**Author responses**
* * *
Review of the revised manuscript "Comparison of ocean vertical mixing schemes in the Max Plank Institute Earth System Model (MPI-ESM1.2)"

I think the revised manuscript is greatly improved as compared to the previous version in terms of clarity. But I still have a few concerns.

I may misunderstand the main purpose of this manuscript. But based on the title and abstract I was expecting to see a more in depth discussion of the differences among the four vertical mixing schemes and how these differences lead to the differences in the simulated mean climate states in MPI-ESM1.2. However, it looks like that the authors would like to present the differences in the simulated mean climate states with the four vertical mixing schemes on a high level without going too deep into the causes of these differences. This is fair as I understand that it is hard to disentangle the climate responses to the changes in parameterizations in a coupled system. But I would suggest the authors to pay more attention on the differences between the four simulations with different ocean vertical mixing schemes, and perhaps a bit less on the common "biases" of these simulations. Otherwise I don't see much added value of this manuscript to the authors' previous publication Gutjahr et al. (2019), which, if I understand it correctly, is an overview of similar MPI-ESM1.2 simulations with a set of different configurations.

We thank the reviewer for his thorough reading and constructive comments. In the following, we answer all raised questions. Before doing so, however, we would like to mention that we are investigating coupled global simulations. The reduction of model errors is a main justification for incorporating new parameterisations into models. We agree that the detailed analysis of new parameterisations is important, but first we are interested in a general overview. In other words, we want to know whether incorporating a more advanced parameterisation is worth the effort, in terms of reduced model errors.

In particular I think there are a few pieces of information missing in the introduction section:
1) What is the main purpose of this study?
We thank the reviewer for raising these questions. We agree that the motivation was not as precisely as it should have been. It is correct that we compare climatic mean states on a high level, because we are interested in the overall effect a change of the vertical ocean mixing has in global coupled simulations. Therefore, we do not (and probably cannot) deduce the effects down to the process level. Such a study would require stand-alone (idealised) simulations, which we think were already published by the developers of the mixing schemes. We underline that investigating such differences in a coupled set-up is challenging because the ocean is coupled to the atmosphere in a complex way so that additional feedbacks need to be considered.

As outlined below, another "added value" is the introduction to MPI_ESM of a new class of mixing schemes (IDEMIX) that includes an improved representation of mixing in the ocean interior.

2) Why are these four schemes chosen, given the many other choices? Or why do you think a comparison among these particular four schemes in MPI-ESM1.2 is helpful?
The choice of comparing PP, KPP and TKE are motivated because these are the most widely used or traditional schemes in CMIP-type coupled models. We are not aware of a more exotic choice in any of the main modelling centres. An unwanted effect all these scheme share is that they rely on an artificial background diffusivity. From the perspective of energy conservation, introducing such an artificial mixing requires energy that is not taken from a reservoir but is spuriously created. This is an unwanted effect in a model. This is where IDEMIX comes in, because it replaces this artificial background diffusivity by a prognostic equation for internal wave energy, whose dissipation then acts as a source for mixing. Although IDEMIX can in principle be connected with PP or KPP, it is natural to combine it with TKE, because both are formulated in terms of energy budgets, which is

necessary to achieve a more consistent mixing scheme. The basic scheme can be extended by other processes such as Langmuir turbulence, lee waves, etc. but the only step forward to a more energetically consistent scheme is by implementing IDEMIX.

3) What do you want to get out of this comparison of the four vertical mixing schemes in MPI-ESM1.2?
There are at least two motivations. First, there was no study before that compares the traditional vertical mixing schemes in MPI-ESM1.2 because only PP has been used until 2019, where also KPP was implemented. Second, since the energy conservation in the ocean is of great importance it was necessary to also implement the TKE scheme and IDEMIX. As with all new implementations it is of interest what effect a different parameterisation has and whether model biases can be reduced. A special focus is laid on the combined TKE+IDEMIX scheme, because from theory it is the most energy conserving scheme of the scheme we compare. Moreover, there is only one other study that studies IDEMIX in a coupled model (Nielsen et al., 2018).

Then, in the presentation and discussions of the simulation results, I would suggest the authors to focus more on differences among these simulations. The present discussions appear a bit unfocused to me.
We think that we focus on comparing the differences between the simulations; however, not at a detailed process level, but at a higher level, because of the intention of the study. We present results of the mean states averaged over the last 20 simulation years (model year 80-100) and focus on model biases. This focus is justified since it is hoped that these biases are reduced or at least it is of interest to see how they are affected by choosing a mixing scheme different from the default option. For some areas, such as the Arctic Ocean/Fram Strait we can link changes in the model bias to the mixing, whereas in other areas, e.g. in Agulhas region, the change in bias results from changes in the circulation. Whenever possible, we try to discuss possible mechanisms, but for a more detailed analysis specific case studies are needed, which is not the aim of our study.

Specific comments:

L41-50: A sentence or two on why the four schemes were chosen given these many more may be helpful. Perhaps moving the paragraph of L29-35 here after this paragraph?
We explain in the text that KPP and TKE are the most widely used vertical mixing schemes in ocean models. In fact, there is not really an alternative to these two. We also explain that PP is still the default option in MPI-ESM1.2 and we therefore include it in our analysis. Of course, there are several extensions and modifications to KPP and TKE (such as Langmuir turbulence), but we consider IDEMIX for two reasons: we want to get rid of artificial background diffusivity and establish an energetically more consistent scheme. These two points are mutually dependent. We have inserted a paragraph at the suggested location to sharpen our motivation and goal we want to achieve with our study.

L64: "ist" -> "is"?
Corrected.

L69: "modified version PP" -> "modified version of PP"?
Corrected.

L82: What is the main difference from the CVMix version? Is the Marsland et al extension the only difference?
Indeed there is another difference we did not mention explicitly, but have added now to the suggested line: "… and in which the diffusivity is independent of the viscosity."

L110: "first" -> "upper"?
Corrected.

L115: Based on Table 1 it looks like you are using PP below the mixed layer for HR_kpp. Does that

mean the only difference between HR_pp and HR_kpp is the treatment in the mixed layer? How is the mixing in the mixed layer treated in HR_pp?

That is correct. The KPP scheme is a pure mixed layer scheme. Therefore, below the mixed layer the parameterisation of the viscosity and diffusivity is independent from KPP. Large et al. (1994) proposed a different formulation to Pacanowski and Philander (1981). However, to be consistent with the PP scheme and with Gutjahr et al. (2019), we use the PP scheme below the mixed layer, or more precise the ocean boundary layer. In the original formulation of Pacanowski and Philander (1981) there is no discrimination between the mixed layer and the interior ocean. That is one reason why the additional mixing induced by wind stress was added to the PP scheme at MPI. So the viscosity and diffusivity in the mixed layer are calculated by equation (A4) and (A5), with the additional term from the wind stress, Kw(z), being active (the term decays exponentially with depth).

L149-150: But it looks like the deep convection in the Labrador Sea is the strongest in HR_ide? What might be the reason?
That is true. It is not straightforward to pinpoint a process that is causing this. It is more that several processes act together to destabilize the water column. A possible reason could be that the inflow of salt from the Mediterranean and the Indian Ocean is strongest in HR_ide. This more saline water in the Atlantic reaches the subpolar gyre, which is reducing the effect of warming on the density with the result of higher density than in the other simulations. An indication for this reasoning is that the mixed layer depths are almost everywhere deeper in the subpolar North Atlantic and Nordic Seas with IDEMIX.

L171: I can see quite some differences in, e.g., the Eastern and Western Equatorial Pacific and the subtropical regions in the Atlantic?

That is correct. We revised the sentence in the manuscript as follows:
"The sea surface salinity is mostly unaffected by the chosen vertical mixing scheme, except in the Arctic Ocean (Fig.3) and in western and eastern equatorial Pacific, which could be related to differences in the feedback with the atmosphere."

L182: The simulations are forced by constant 1950s conditions whereas the PIOMAS represent the average of 1979-2005. Do you expect they will match each other if the model is perfect?
No, we do not expect to match PIOMAS with 1950s conditions. We have added the PIOMAS ice thickness to illustrate how the thickness distribution looks like in the (more) recent past. Since there is no other comparable data set for sea ice thickness, we think it is a justified comparison. At least from the perspective of climate warming, we would not expect thinner ice than in PIOMAS for 1950s conditions.

L184-186: Are you comparing with HR_pp? HR_ide seems to have an ice edge to the further north.
Yes, we compare with HRpp.

L188: By TKE here are you referring to both HR_tke and HR_ide?
Yes we do, but we have reformulated the sentence to make that clear: "The more northerly location of the ice edge in the KPP and TKE(+IDEMIX) simulations … ." We also made this clear in the sentence before and after this one.

L204-205: Comparing which vertical mixing schemes?
We refer here to TKE+IDEMIX: "…, for instance with TKE+IDEMIX the warm bias is reduced in the Arctic Ocean but enhanced for the MOW (Fig.6)."

L206: Fig 8 introduced before Fig 7?
Thank you for spotting this error. We have switched the position of Fig.7 and Fig.8. Now the salinity bias map is referred to before the inflow from the Indian Ocean.

L211-212: What about the salinity bias in the South Pacific?
That is correct; we did not mention this bias here. We have added a sentence to the paragraph:
"We further notice a large salinity bias in the South Pacific that is linked to the South Pacific gyre.

We speculate that the bias is due to unresolved eddies in the East Australian Current and processes at the border zone of the South Pacific Current and the Subantarctic Front. A more detailed analysis is needed to identify the cause of this model bias, which we suspect to be related to model resolution but will not explore here."

==L213: Is the streamfunction in Fig 7 integrated in the vertical over the full depth, or from the surface to some depth?==
It is integrated over the full depth. We have added "vertically integrated" to the caption of now Fig. 8.

==L219-220: You can probably see this by looking at vertical sections of the salinity in this region.==

We have plotted a vertical cross-section along 36°N in the Atlantic showing the distribution of salinity in HRpp and the differences of the KPP and TKE(+IDEMIX) simulations. This figure shows higher salinity already in the Mediterranean Sea with KPP and TKE+IDEMIX (Fig.1b and d). At a depth of about 1000 m, where the overflows are in equilibrium with the ambient water, a clear imprint of the higher salinity can be seen in all three sensitivity simulations, but it is most pronounced in TKE+IDEMIX (Fig.1d). This result indicates that IDEMIX influences the watermass properties in the Mediterranean Sea and the mixing of the overflow plume.

We revised the relevant paragraph in the manuscript to: "However, the MOW is already saltier by about 0.4psu and 0.5°C warmer before it flows into the Atlantic. It remains a subject of further investigation what causes this warmer and saltier MOW in TKE+IDEMIX. Possibly, it modifies the variability of the near-surface wind field or the net evaporation over the Mediterranean Sea (Aldama-Campino and Döös, 2020)."

[Figure]

*Figure 1: Vertical cross-section along 36°N in the Atlantic Ocean (averaged over last 20 model years) showing (a) absolute potential temperature in HRpp, and (b-d) the differences "experiment" minus HRpp.*

L229-231: This reflects one of the issues in this set of simulations. If you run the simulations (forced by constant 1950s conditions) for longer, you might see different patterns of the "biases", right? So how sensible is the magnitude of the "biases" shown here in these figures?
Yes, it is possible that the bias pattern changes over longer integration periods. However, because we follow the HighResMIP protocol, we only conducted 100-year long simulations with these model configurations. Therefore we wrote at the end of section 3: "The simulation length of 100 years is too short to see pronounced effects in the deep ocean, but it could be expected that over longer periods (several centuries) the additional mixing from internal waves might affect the diapycnal diffusion of the upwelling deep water, e.g. in the Pacific." Thus, the biases and their differences between the simulations stem from quick (100years) responses. On longer time scales, also the slower reacting deep ocean becomes involved.

L233: Delete "of"?
Corrected.

L234-237: I'm confused, are you discussing Fig 7 in this paragraph?
Yes, we refer to Fig.7 (now Fig.8)

L245-248: This paragraph is confusing.
We have revised the paragraph to: "In the Nordic Seas the water temperature is higher in all simulations with KPP and TKE(+IDEMIX) than in HRpp. Although the higher temperature partly compensates for the increase in salinity, the overflows across the Greenland-Scotland Ridge are dense enough and reach depths of about 3000m. Their warmer temperature can be seen at and south of 60°N. Similarly, also the deep water formed in the Labrador Sea is warmer than in HRpp and together these water masses cause a warm bias when exported as the Deep Western Boundary Current."

L258-260: I don't understand this sentence.
We have revised the sentence to: "To compensate for the increased overturning in the Nordic Seas, the water in the Atlantic must be replaced by a stronger inflow from the Indian Ocean, which is the case in the simulations with KPP and TKE(+IDEMIX), as seen in Fig.8".

L361: Do you mean "warm halocline layer"?
In addition, brine rejection in the interior Arctic is less effective as a mixing mechanism because of the strong stabilising vertical salinity gradient.

L436: Delete "but"?
We revised the sentence to "Although not observed by Argo floats, all simulations show deeper mixed layers north of 50°S in the Pacific Ocean east of New Zealand and in the South Atlantic."

L438-439: What do you mean by "in addition, we compare them with control simulations"?
What we meant is that we compare simulations forced by constant 1950s GHG with ARGO observations of the recent past. We revised the sentence to: "It should be kept in mind, however, that comparing model simulations with Argo float data is always difficult because the floats do not measure continuously in time and space. Moreover, we compare the Argo float data of the recent past with simulations driven by a constant greenhouse gas concentration from 1950."

Fig 1: This figure shows the differences between the four vertical mixing schemes in the ocean interior very nicely. I wonder if it is worth to show their differences in the mixed layer as well, which may be more relevant to the ocean surface states such as SST, SSS and MLD.
Probably yes. However, we tried to find a trade-off between the number of figures and the information we want to show. We decided to have 2d maps of SST, SSS and MLD in the manuscript because these are important fields for the coupling, and furthermore readers are interested in these fields. We agree that a more detailed analysis of the mixed layer would be of interest in a future study.

References

Gutjahr, O., Putrasahan, D., Lohmann, K., Jungclaus, J. H., von Storch, J.-S., Brüggemann, N., Haak, H., and Stössel, A.: Max Planck Institute Earth System Model (MPI-ESM1.2) for the High-Resolution Model Intercomparison Project (HighResMIP), Geosci. Model Dev., 12, 3241–3281, https://doi.org/10.5194/gmd-12-3241-2019, 2019.
* * *
Review #2 of:" Comparison of ocean vertical mixing schemes in the Max Planck Institute Earth System Model (MPI-ESM1.2)." by Oliver Gutjahr et al.

The paper compares the effects of four different ocean vertical mixing schemes provided by the Community Vertical Mixing (CVMIX) library on the mean state simulated by the coupled model MPI-ESM1.2. Used are the Pacanowski and Philander (PP), K-profile (KPP), turbulent kinetic energy (TKE) vertical mixing schemes as well as a prognostic schemes for internal wave energy and its dissipation (IDEMIX) which is combined with TKE. The author addresses temperature and salinity biases in a global and regional context with sufficient analysis of there causes and impacts. Compared to the first revision round the paper has significantly improved in scientific impact but also in its phrasing and expression. All comments of the reviewer have been considered. The paper was restructured for a better understanding and new chapters for the impact of the mixing schemes on the sea ice but also on the coupled atmosphere have been added. These chapters are of particular value for coupled modeling community.

We thank reviewer 2 for his time and positive feedback.

From my point their are only minor things to change (linenumber with respect to file gmd-2020-202-ATC1.pdf):

line 67: "...submodel JSBACH3.2, and of the ice-ocean...", delete of
Corrected.

line 73: "...River runoff ist calculated...", exchange to "is"
Corrected.

line 79: "...is a modified version PP scheme..." exchange with "...is a modified version of the PP scheme..."
Corrected.

line 110: "...wave-wave interaction is accounted for that might...", please reformulate
We rephrased the sentence to: "When internal waves propagate, they can be damped by wave-wave interaction, which is taken into account in IDEMIX".

line 153: "...approximately homogeneously distributed to leading order...", not fully clear what the author want to say here
We rephrased this sentence to: "Figure 1 shows the spatial distribution of the vertical diffusivity K for the model layer at 1020m depth. Apart from boundary flows, deep convection regions and the surface mixed layer, K is approximately homogeneous in the simulations with PP and KPP. Both simulations use a simple constant background diffusivity of $K=1.05*10^{-5}$ ($m^2$ $s^{-2}$) to parameterise internal wave breaking."

line 203: "...Ice thickness is lower in all simulations than in PIOMAS..." exchange with "...Ice thickness is lower in all simulations compared to PIOMAS..."
Corrected.

line 226: "...where they affect to local...", exchange with "...where they affect the local..."
Corrected.

line 278: "...across the Greenland-Scotland Ridge at and ...", delete at
We revised the sentence due to a comment from reviewer 1 to: "In the Nordic Seas the water temperature is higher in all simulations with KPP and TKE(+IDEMIX) than in HRpp. Although the higher temperature partly compensates for the increase in salinity, the overflows across the Greenland-Scotland Ridge are dense enough and reach depths of about 3000m. Their warmer temperature can be seen at and south of 60°N. Similarly, also the deep water formed in the

Labrador Sea is warmer than in HRpp and together these water masses cause a warm bias when exported as the Deep Western Boundary Current."